# CoT-PL: Visual Chain-of-Thought Reasoning Meets Pseudo-Labeling for Open-Vocabulary Object Detection

## Abstract

Open-vocabulary object detection (OVD) seeks to recognize and localize object categories beyond those seen during training. Recent approaches typically leverage vision-language models (VLMs) to generate pseudo-labels using image-text alignment, allowing detectors to generalize to unseen classes without explicit supervision. However, these methods depend heavily on direct image–text matching, neglecting the intermediate reasoning steps essential for interpreting semantically complex scenes. This results in limited robustness when confronted with crowded or occluded visual contexts. In this paper, we introduce CoT-PL, a new framework that employs structured visual chain-of-thought (CoT) reasoning into the pseudo-labeling process. CoT-PL decomposes object understanding into three interpretable steps: (1) region perception even for unseen objects, (2) category recognition via zero-shot reasoning, and (3) background grounding to separate semantically complex objects. Crucially, the third step naturally motivates our contrastive background learning (CBL) that uses the pre-computed background cues as negatives to promote feature disentanglement between objects and background. In this way, CoT reasoning and CBL form an integrated pipeline tailored to robust pseudo-labeling in crowded or occluded scenes. Notably, in these two settings, our novel-class pseudo-label quality achieves relative improvements of 103.4% and 168.4% over the best prior, respectively. Our extensive experiments demonstrate that CoT-PL achieves +7.7 $AP_{50}$ on open-vocabulary COCO and +2.9 mask AP on LVIS for novel classes, setting a new state of the art.

## 1 Introduction

Open-vocabulary object detection (OVD) aims to localize both seen (base) and unseen (novel) categories at test time, using only base-class annotations during training. To bridge this supervision gap between seen and unseen categories, recent approaches leverage vision-language models (VLMs) pre-trained on large-scale image-text pairs (Radford et al., 2021). These VLMs map textual descriptions to visual representations, allowing OVD methods to recognize novel classes.

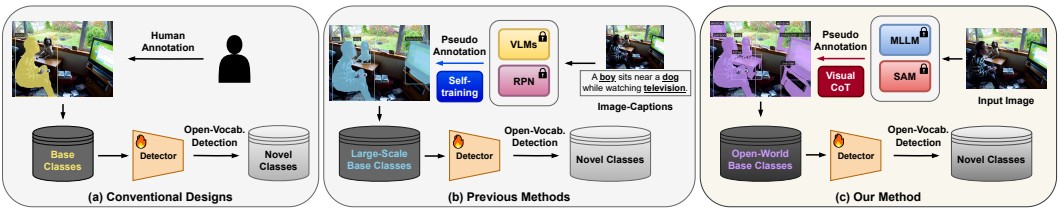

Figure 1: **Trends in pseudo-labeling for OVD. (a)** Manual pseudo-labels for novel classes are costly and do not scale. **(b)** Recent self-training methods automate pseudo-labeling by labeling region proposals via similarity with category text embeddings using vision-language models (VLMs), but degrade with VLMs' poor object localization and caption-dependent vocabulary. **(c)** We employ visual chain-of-thought with Segment Anything Model (Kirillov et al., 2023) and multimodal large language models (Bai et al., 2023) for accurate object perception and zero-shot category recognition.

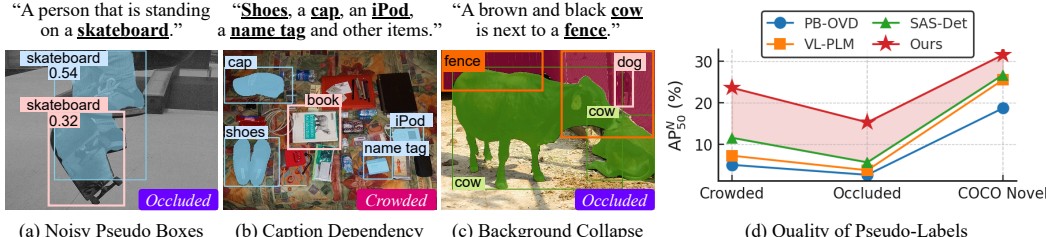

(a) Noisy Pseudo Boxes    (b) Caption Dependency    (c) Background Collapse    (d) Quality of Pseudo-Labels

Figure 2: **Limitations and quality of pseudo-labels for complex scenes. (a)** Noisy pseudo boxes due to poor object localization by VLMs, **(b)** limited object coverage from captions, and **(c)** occluded objects treated as background. **(d)** Pseudo-label quality on the OV-COCO validation set: *Crowded* denotes images with many objects and *Occluded* denotes objects occluded by other objects.

Among such efforts, pseudo-labeling has emerged as a state-of-the-art approach for OVD by augmenting the base set with automatically generated annotations that partially cover novel classes (Gao et al., 2022; Pham et al., 2024). Early pseudo-labeling methods for OVD relied on manual annotation of novel classes, which was costly and lacked scalability (Figure 1-a). More recent approaches (Zhao et al., 2022; 2024) automate pipelines that generate pseudo-annotations for novel classes (Figure 1-b). This process involves generating pseudo-labels for region proposals during self-training using CLIP (Radford et al., 2021), based on the similarity between region features and text embeddings of potential object categories, typically derived from dataset class names or image captions.

Despite their strong performances in general scenes, state-of-the-art OVD approaches still struggle in challenging scenarios like crowded or occluded objects. A key reason is their reliance on direct image–text matching via CLIP, lacking the intermediate visual reasoning steps required for understanding complex scenes (Yüksekgönül et al., 2023). Recent pseudo-labeling approaches also inherit the same limitation. We identify three key factors underlying their failure in complex settings. **(L1) Noisy pseudo boxes**: VLMs trained with image-level supervision exhibit co-occurrence bias, favoring crops with semantically related objects (Zhong et al., 2022). This inductive bias hinders object-level pseudo-box labeling in complex, occluded scenes. In Figure 2-a, CLIP incorrectly assigns the highest similarity to the token "skateboard" for a pseudo box with the person's feet partially hidden behind the skateboard. **(L2) Caption dependency**: Captions lack detail particularly in complex, crowded scenes. As a result, objects not listed in captions remain unlabeled. Figure 2-b shows missing pseudo annotations for many stuff categories (*e.g.,* book) which are absent from the caption. **(L3) Background collapse**: Detecting objects under occlusion is challenging. These instances are frequently unlabeled and learned as background in training (Li et al., 2024). As illustrated in Figure 2-c, a dog partially occluded by a fence was not detected and treated as background.

To directly tackle these three limitations associated with complex scenes, we argue that pseudo-labeling for OVD must be reformulated as an interpretable multi-step reasoning process rather than a single-step alignment. In the vision-language domain, visual chain-of-thought (CoT) prompting enhances VLM reasoning by encouraging step-by-step thinking (Wu et al., 2023a). Inspired by this, we design pseudo-label generation using a three-step CoT pipeline explicitly aligned with OVD challenges: (1) region perception with SAM (Kirillov et al., 2023) produces candidate masks, and a multimodal large language model (MLLM) verifies object existence to filter out spurious or partial boxes, mitigating (L1); (2) zero-shot category recognition assigns labels to each region without relying on captions, eliminating (L2); and (3) contextual background grounding distinguishes true objects from unlabeled background regions, resolving (L3). Importantly, the third step naturally leads to our contrastive background learning (CBL), which transforms grounded background cues into negative signals for training. This integration does more than relying on strong models because SAM and MLLMs alone produce noisy or inconsistent labels but our structured reasoning and background-aware learning transform their raw outputs into high-quality pseudo-labels. In this way, CoT reasoning and CBL form a single, integrated pipeline purpose-built for robust pseudo-labeling in complex scenes. In Figure 2-d, our method significantly outperforms prior works (Gao et al., 2022; Zhao et al., 2022; 2024) in generating pseudo-labels under challenging environments.

We conduct extensive experiments on two OVD benchmarks, OV-COCO (Lin et al., 2014) and OV-LVIS (Gupta et al., 2019). Under challenging conditions like crowding and occlusion, our method

demonstrates superior pseudo-label quality compared to prior state-of-the-art pseudo-labeling methods. Our method sets a new state of the art, improving box $AP_{50}$ for novel classes on OV-COCO by 7.7 and mask mAP on OV-LVIS by 2.9, compared to prior work (Wu et al., 2023c). We report pseudo-label statistics across OVD benchmarks, and our key contributions include:

- To our knowledge, we are the first to reformulate pseudo-labeling in OVD as a visual chain-of-thought, decomposing complex-scene object understanding into an interpretable multi-step reasoning process beyond single-step VLM alignment.
- We introduce CoT-PL, a unified system for robust pseudo-labeling in complex scenes by integrating CoT reasoning and contrastive background learning (CBL) that uses grounded background cues as negative training signals.
- Performance steadily improves with stronger teacher MLLMs, achieving state-of-the-art results in OVD with high-quality pseudo-labels, even in challenging environments.

## 2 RELATED WORK

### 2.1 CHAIN-OF-THOUGHT (COT) REASONING

Chain-of-thought (CoT) reasoning has emerged as a powerful approach in natural language processing, enabling models to tackle complex reasoning tasks by incrementally decomposing them into interpretable steps. Initial work (Wei et al., 2022) demonstrated that large language models produced more accurate outcomes by generating intermediate reasoning before arriving at a final answer. In the visual domain, multimodal chain-of-thought methods process visual inputs sequentially to reason about future states. These approaches have been applied to diverse tasks, including bounding box prediction (Shao et al., 2024), planning in autonomous driving (Tian et al., 2024), intermediate image infillments (Rose et al., 2023), and CLIP embedding synthesis (Harvey & Wood, 2023). In the vision-language-action setting, CoT reasoning has recently gained traction for guiding closed-loop robotic manipulation through sub-goal images as intermediate reasoning steps (Zhao et al., 2025). In this work, we extend visual chain-of-thought reasoning to generate high-quality pseudo-labels for open-vocabulary object detection, even in semantically complex scenes.

### 2.2 OPEN-VOCABULARY OBJECT DETECTION

Open-vocabulary object detection (OVD) aims to detect novel objects not seen during training by leveraging vision-language models (VLMs) (Radford et al., 2021) trained on large-scale image-text pairs. Recent OVD methods (Du et al., 2022; Wu et al., 2023d) employ prompt modeling to transfer knowledge through learned prompts, enabling more precise contextual descriptions of each class. Several studies (Gu et al., 2022; Wu et al., 2023c) use knowledge distillation to align detectors with VLM features for recognizing unseen objects. Other approaches (Jin et al., 2024; Liu et al., 2024a) reinforce the text modality using large language models (LLMs). Meanwhile, InstaGen (Feng et al., 2024) focuses on the image modality, improving novel class prediction via synthetic images from an image generation model. Furthermore, Grounding DINO (Liu et al., 2024b) introduces prompt-based object detection by facilitating cross-modal information exchange between VLMs and transformers. Another OVD approach is pseudo-labeling, which addresses the limited base classes by leveraging extended supervision. These approaches often generate pseudo-annotations or weak supervision through self-training using image captions (Gao et al., 2022; Zhao et al., 2022; 2024) or dataset class names (Zhao et al., 2024). However, they focus on direct image–text matching via CLIP, disregarding a reasoning process necessary for complex scenes (Yüksekgönül et al., 2023). We reformulate OVD pseudo-labeling as a sequence of interpretable reasoning steps using visual chain-of-thought (CoT), thereby enabling robust pseudo-labeling in challenging environments.

## 3 METHODOLOGY

We introduce CoT-PL, a structured visual chain-of-thought pipeline tailored to robust pseudo-labeling, even in two challenging scenarios. Captions in crowded scenes are underspecified, while simple CLIP matching lacks fine-grained visual perception under occlusion. To address these issues, CoT-PL recasts object understanding as three interpretable reasoning steps: (1) (Pseudo Box

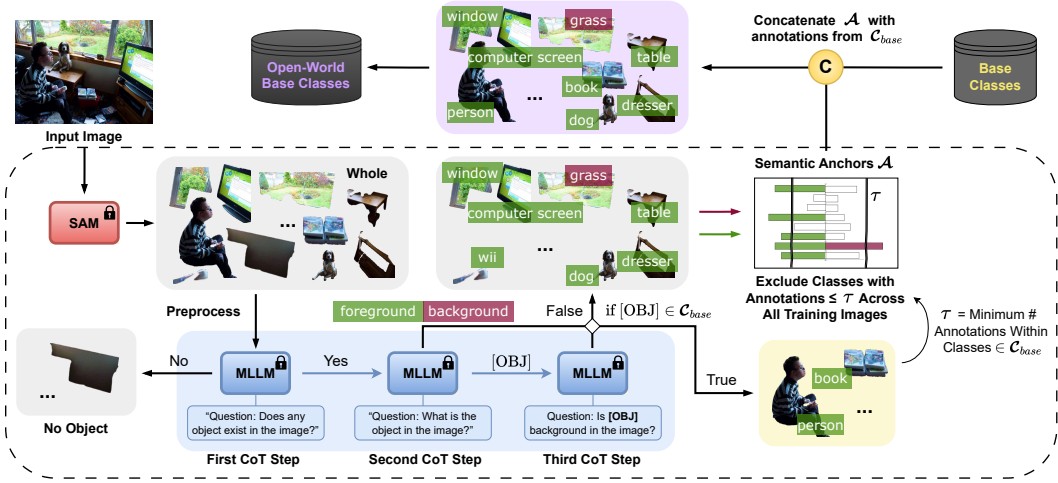

Figure 3: **Overview of the proposed visual chain-of-thought pipeline.** Our method first queries an MLLM about object existence inside SAM-generated pseudo boxes on preprocessed images, then performs zero-shot labeling, and finally extracts background representations nearby. Refined with semantic anchors for reliability, the resulting pseudo-annotations are merged into the base set.

Generation) We generate boxes for all instances using SAM (Kirillov et al., 2023) and verify if each box corresponds to a valid object. (2) (Pseudo Label Assignment) We assign a pseudo-label to each box through zero-shot object recognition using an multimodal large language model (MLLM). (3) (Background Extraction) We extract background representations by leveraging the MLLM's grounding capability. Specially, the third step naturally leads to a contrastive background learning (CBL) strategy, which uses the identified background concepts as negative training signals to promote feature disentanglement between objects and background.

## 3.1 PSEUDO BOX GENERATION

**First CoT step.** We aim to generate accurate pseudo-bounding boxes for all potential objects in the train set. To this end, we leverage the strong generalization ability of SAM to produce segmentation masks for all object instances in each image. Relying on low-level visual cues such as edges and color contrasts, SAM is widely used in open-vocabulary settings to generalize beyond base classes to unseen objects (Zhang et al., 2023; Qin et al., 2024). However, applying class-agnostic SAM to OVD presents two key challenges: (1) it segments regions at varying semantic granularity (*i.e.* full objects vs. parts), and (2) it may include non-object regions such as background.

To address (1), we follow a similar approach to recent work (Qin et al., 2024), which utilizes SAM to extract accurate object-level masks at multiple semantic levels (*i.e.* whole, part, sub-part). As illustrated in Figure 3, our method selects whole-instance masks to generate precise object-level pseudo boxes that tightly enclose them. To address (2), we then use the zero-shot reasoning capability of a robust MLLM (Bai et al., 2023) to verify whether each box contains a valid object. For example, we prompt the model with: "`Question: Does any object exist in the image?`" and interpret the response ("Yes" or "No" or "Unsure") as a ternary classification of object existence. Boxes classified as "No" or "Unsure" are discarded, whereas those classified as "Yes" are passed to the next stage of the CoT pipeline. For example, in Figure 3, regions lacking discernible objects (*i.e.* plain dark areas) are discarded. For "Unsure" responses—a rich source of long-tailed cases, we record the model's rationale to support explainability. At this stage, we achieve region perception by generating accurate pseudo boxes that tightly enclose identifiable yet unlabeled objects.

## 3.2 PSEUDO LABEL ASSIGNMENT

**Second CoT step.** Most OVD methods rely on CLIP for pseudo-labeling, but it lacks fine-grained visual perception and requires predefined class names (Yüksekgönül et al., 2023). These limitations inevitably lead to inaccurate or missing pseudo-labels, especially in complex scenes. This motivates

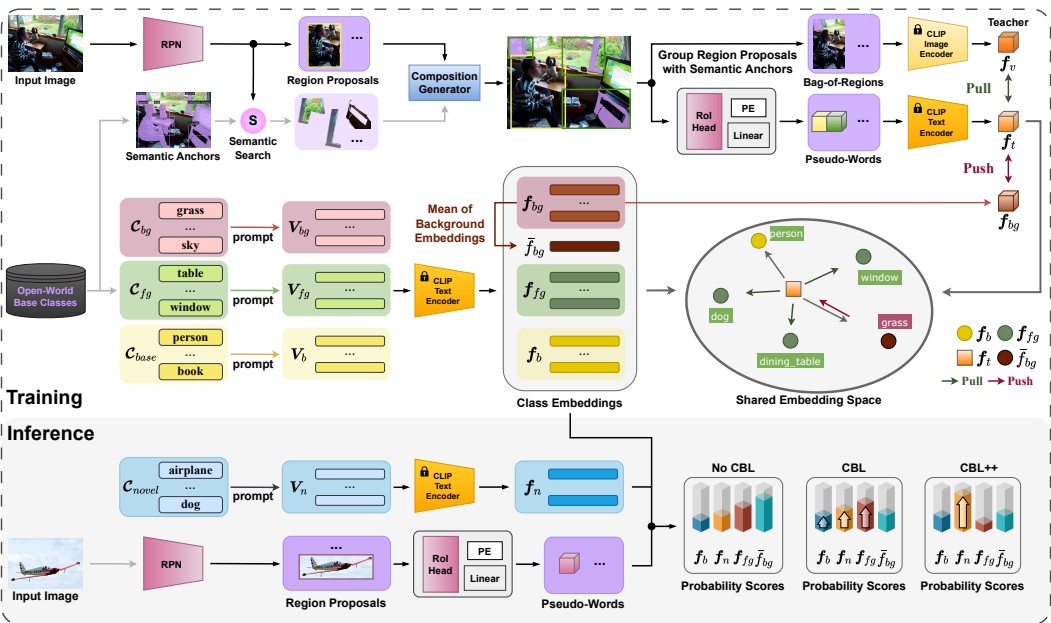

Figure 4: **Overall architecture of CoT-PL.** Built on BARON (Wu et al., 2023c) (See Appendix G), the proposed method encodes the open-world base set, partially including novel classes, using the CLIP text encoder. The CLIP embeddings of multiple background concepts are averaged to initialize a single learnable background embedding. These concepts are also used as negative samples in contrastive learning to encourage feature disentanglement between objects and background. At inference time, we apply CBL++ to mitigate class interference by removing pseudo-labels associated with the ground-truth novel classes.

the second CoT step, which assigns more accurate pseudo-labels to each box independently of image captions. To achieve this, we leverage an MLLM with strong zero-shot multi-class identification to infer the object category within each box (Wang et al., 2023a). As shown in Figure 3, during the second CoT stage, we prompt "Question: What is the object in the image?", expecting a specific class label beyond the base set. This simple query lays the groundwork for caption-free, open-world pseudo-labeling in OVD.

However, MLLMs are sensitive to visual content and often produce irrelevant predictions when queried on pseudo boxes, as attention may leak beyond the target region (Zang et al., 2025; Zhang et al., 2025b; Fu et al., 2024). A common workaround applies hard masks to remove non-target regions, but this often induces errors from misleading visual content (Chang et al., 2023; Fontanini et al., 2023). For example, a masked giraffe's silhouette may bias the model to misclassify a tree as a giraffe (See Appendix H.1). To mitigate this, we apply grayscaling and blurring as a soft mask to suppress non-target regions, helping the MLLM focus and produce more accurate pseudo labels (Yang et al., 2023). This preprocessing step is visualized in Appendix 6.

To further improve pseudo-label reliability, we add a post-processing step that filters out outlier labels, chiefly reducing false positives. We exclude pseudo-labels with fewer annotation counts than a threshold, as they are likely unreliable. For simplicity, the threshold is set to the minimum annotation count among base classes. The remaining high-confidence labels, or *semantic anchors*, are then integrated with the base class set to construct our open-world base set. Figure 3 highlights anchors with color bars: green (*foreground*), red (*background*), and white (*outlier*). Without predefined class names or image captions, the second CoT step yields robust and reliable pseudo-labels by integrating MLLM zero-shot reasoning with reliability-enhancing pre- and post-processing.

## 3.3 BACKGROUND EXTRACTION

**Third CoT step.** While stronger than VLM-based methods, our MLLM-based pseudo-labeling is not yet complete for long-tailed complex scenes because it hinges on MLLM capability. In such

cases, the model often returns "Unsure" for small or occluded objects (*i.e.* a mug hidden behind a laptop). For example, we observe that weaker models produce more "Unsure" responses and fewer valid labels across diverse objects (See Table 2). These regions remain unlabeled and are treated as background during training (Bansal et al., 2018; Li et al., 2024), since they match neither base nor pseudo-labeled classes, a phenomenon we call *background collapse*.

Unfortunately, explicitly identifying unlabeled, inherently unknown objects is nontrivial. Instead, we sidestep this by disentangling collapsed objects from background representations in the feature space. To this end, we use MLLM grounding as our third CoT step to verify whether a given "`[OBJ]`" belongs to background via a binary prompt ("Yes" or "No"). This simple binary query helps determine whether a given object category is background. For example, the model classifies "grass" as background and "drawer" as foreground (See Figure 3). These grounded background cues serve as negative supervision, encouraging object–background disentanglement in training.

### 3.4 CONTRASTIVE BACKGROUND LEARNING (CBL)

Our CBL instantiates the idea on BARON (Wu et al., 2023c) that performs competitively through online sampling of compositional structures (*i.e.,* co-occurrence of objects) as training signals. However, the sampling process is computationally expensive and time-consuming. In Figure 4, we design a composition generator that groups cached, pre-computed semantic anchors for each proposal into a *bag of regions*, reducing overall training time by $4\times$. Then, these sampled regions are projected into the word embedding space using the linear layer within Faster R-CNN (Ren et al., 2015), resulting in pseudo-words. The pseudo-words are passed through the text encoder $\mathcal{T}$ to obtain the bag-of-regions embedding $f_t^i = \mathcal{T}(w_0^i + p_0^i, w_1^i + p_1^i, \cdots, w_{N^i-1}^i + p_{N^i-1}^i)$, where $N^i$ is the number of regions in the $i$-th bag, $p_j^i$ represents the positional embedding of the $j$-th region in the $i$-th bag. Finally, this bag-of-regions embedding is aligned with the VLM's image embeddings $f_v^i = \mathcal{V}(b_0^i, b_1^i, \cdots, b_{N_i}^i)$, where $b_j^i$ is the $j$-th region in the $i$-th bag. Further implementation details appear in Appendix G.

To alleviate the background collapse, we propose a contrastive background learning (CBL) strategy that explicitly disentangles objects, including unlabeled objects, from true background representations (*i.e.* grass or sky) in the feature space during training. As shown in Figure 4, we first encode the base categories $\mathcal{C}_{base}$, pseudo-labels $\mathcal{C}_{fg}$, and identified background concepts $\mathcal{C}_{bg}$ using the CLIP text encoder with the prompt template "`a photo of [OBJ]`". The averaged background embedding $\bar{f}_{bg}$ serves as an initialization for a learnable background prior. To encourage feature discrimination, we apply a contrastive objective in which the background embeddings $f_{bg}$ serve as negative samples, formulated as the alignment InfoNCE loss (Rusak et al., 2024):

$$\mathcal{L}_{\text{bag}} = \frac{1}{2} \sum_{k=0}^{G-1} \left( \log p_{t,v}^k + \log p_{v,t}^k \right), \tag{1}$$

where $G$ is the number of bags, and the $p_{t,v}^k$ and $p_{v,t}^k$ can be calculated as:

$$p_{t,v}^k = \frac{\exp(\tau' \cdot \langle f_t^k, f_v^k \rangle)}{\sum_{l=0}^{G-1} \exp(\tau' \cdot \langle f_t^k, f_v^l \rangle) + \sum_{j=0}^{M-1} \exp(\tau'' \cdot \langle f_t^k, f_{bg}^j \rangle)}, \tag{2}$$

$$p_{v,t}^k = \frac{\exp(\tau' \cdot \langle f_v^k, f_t^k \rangle)}{\sum_{l=0}^{G-1} \exp(\tau' \cdot \langle f_v^k, f_t^l \rangle) + \sum_{j=0}^{M-1} \exp(\tau'' \cdot \langle f_v^k, f_{bg}^j \rangle)}, \tag{3}$$

where $\langle \cdot, \cdot \rangle$ denotes cosine similarity; $M$ is the number of background concepts; $\tau'$ and $\tau''$ are temperature scaling factors. The loss encourages alignment between matched text–image pairs (*e.g.,* dog and window) while pushing apart mismatched background representations (*e.g.,* grass).

## 4 EXPERIMENTS

**Datasets and evaluation metrics.** We evaluate our CoT-PL using two widely used OVD datasets: OV-COCO (Lin et al., 2014) and OV-LVIS (Gupta et al., 2019). We adopt the category split approach from OVR-CNN (Zareian et al., 2021) for the OV-COCO, dividing object categories into 48 base and 17 novel categories. For OV-LVIS, we follow ViLD Gu et al. (2022), separating the 337 rare

Table 1: **Result comparisons on OV-COCO** (Lin et al., 2014). Methods are categorized into two groups based on whether additional supervision beyond instance-level labels in the base classes $\mathcal{C}_B$ is utilized during training, *i.e.*, weak or pseudo labels. Note that $\mathcal{C}_N$ is the novel class set; RN50-C4 uses features from the fourth convolutional stage, and RN50-FPN uses a multi-scale feature pyramid.

| Methods | Supervisions | Backbone | $AP_{50}^N$ (%) | $AP_{50}^B$ (%) |
|---|---|---|---|---|
| **Annotation:** Extra caption datasets, Weak/Pseudo Labels in $\mathcal{C}_B \cup \mathcal{C}_N$ | | | | |
| Detic (Zhou et al.) | ImageNet21K & CC3M | RN50-C4 (24M) | 27.8 | 42.0 |
| OV-DETR (Zang et al.) | Pseudo annotation | RN50 (24M) | 29.4 | 52.7 |
| CoDet (Ma et al.) | CC3M & COCO Caption | RN50 (24M) | 30.6 | 46.4 |
| PB-OVD (Gao et al.) | COCO Caption | RN50-C4 (24M) | 30.8 | 46.4 |
| VL-PLM (Zhao et al.) | Pseudo instance-level annotation | RN50 (24M) | 34.4 | 60.2 |
| RegionCLIP (Zhong et al.) | CC3M | RN50-C4 (24M) | 35.2 | 57.6 |
| OC-OVD (Rasheed et al.) | COCO Caption | RN50-FPN (24M) | 36.6 | 49.4 |
| SAS-DET (Zhao et al.) | COCO Caption | RN50-C4 (24M) | 37.4 | 58.5 |
| **CoT-PL (Ours)** | Pseudo annotation | RN50-FPN (24M) | **41.7** | 59.4 |
| **Annotation:** Instance-level labels in $\mathcal{C}_B$ | | | | |
| ViLD-ens (Gu et al.) | CLIP | RN50-FPN (24M) | 27.6 | 51.3 |
| BARON (Wu et al.) | CLIP | RN50-FPN (24M) | 34.0 | 60.4 |
| CFM-ViT (Kim et al.) | CLIP | ViT-L/16 (307M) | 34.3 | 46.4 |
| CORA (Wu et al.) | CLIP | RN50 (24M) | 35.1 | 35.4 |
| BIND (Zhang et al.) | CLIP | ViT-B/16 (86M) | 36.3 | 50.2 |
| CLIP-Self (Wu et al.) | CLIP | ViT-B/16 (86M) | 37.6 | - |
| LBP (Li et al.) | CLIP | RN50-FPN (24M) | 37.8 | 58.7 |
| CCKT-Det (Zhang et al.) | CLIP | RN50 (24M) | 38.0 | - |
| OV-DQUO (Wang et al.) | CLIP | RN50 (24M) | 39.2 | - |

Table 2: **Statistics of pseudo labels.** We report the number of pseudo class labels and annotations obtained from 118,287 training images across open-vocabulary benchmarks. The class count indicates how many classes in each benchmark are covered by pseudo-labels.

| Model | Total | | | OV-COCO (65 classes) | | OV-LVIS (1,203 classes) | |
|---|---|---|---|---|---|---|---|
| | # Classes | # Annotations | # Unsure | # Classes | # Annotations | # Classes | # Annotations |
| BLIP2 (2023) | 6,036 | 395,052 | 1,528,251 | 31 | 197,086 | 105 | 137,353 |
| InstructBLIP (2023) | 3,106 | 566,619 | 1,132,179 | 30 | 379,528 | 93 | 315,346 |
| Qwen2 (2023) | 3,916 | 637,349 | 563,156 | 65 | 349,399 | 115 | 232,298 |

categories as novel and consolidating the common and frequent categories into base categories. We follow OVR-CNN for evaluation: on OV-COCO, we report box AP at IoU 0.5 for novel categories ($AP_{50}^N$); on OV-LVIS, we report mask mAP over IoUs from 0.5 to 0.95 for rare categories ($AP_r$).

**Implementation details.** We build CoT-PL on Faster R-CNN (Ren et al., 2015) with ResNet50-FPN. For a fair comparison, we initialize the backbone network with weights pre-trained by SOCO (Wei et al., 2021) and use synchronized Batch Normalization Zhang et al. (2018), following recent studies Wu et al. (2023c); Du et al. (2022). For the main experiments on OV-COCO and OV-LVIS, we choose the $1\times$ and $2\times$ schedules, respectively. We use the CLIP model based on ViT-B-16 (Dosovitskiy et al., 2021) as our pre-trained VLM. For the prompt of category names, we default to the hand-crafted prompts from ViLD (Gu et al., 2022) in all our experiments on OV-COCO and OV-LVIS. We follow the same hyperparameter settings as the baseline (See Appendix E).

## 4.1 MAIN RESULTS

**Comparison with state-of-the-art methods.** Most recent OVD methods utilize weak or pseudo-annotations during training. For instance, RegionCLIP (Gao et al., 2022) and CoDet (Ma et al., 2023) utilize additional caption datasets to discover novel concepts, while OV-DETR (Zang et al., 2022) further generates pseudo-annotations in a self-training manner. As shown in Table 1, on OV-COCO, our CoT-PL achieves the best performance ($AP_{50}^N$ 41.7%) among methods using additional supervision, with the default ResNet50 (He et al., 2016) backbone. Furthermore, CoT-PL surpasses

Table 3: **Result comparisons on OV-LVIS** (2019). Our CoT-PL achieves competitive performance on instance segmentation.

| Methods | $AP_r$ (%) | AP (%) |
|---|---|---|
| ViLD-ens (Gu et al., 2022) | 16.6 | 25.5 |
| Detic (Zhou et al., 2022) | 17.8 | 26.8 |
| BARON (Wu et al., 2023c) | 19.2 | 26.5 |
| MIC (Wang et al., 2023c) | 20.8 | 30.7 |
| OADP (Wang et al., 2023b) | 21.7 | 26.6 |
| LBP (Li et al., 2024) | 22.2 | 29.1 |
| **CoT-PL (Ours)** | **23.0** | 29.4 |

Table 4: **Pseudo-label quality on OV-COCO validation**. Metrics on novel-class pseudo-labels. *Crowded* denotes images with more objects than the average in this dataset. *Occluded* denotes novel-class objects over 50% covered by other ground-truth boxes.

| Methods | COCO Novel | *Crowded* | *Occluded* |
|---|---|---|---|
| PB-OVD | 18.7 | 5.1 | 2.7 |
| VL-PLM | 25.5 | 7.3 | 3.8 |
| SAS-Det | 26.7 | 11.6 | 5.7 |
| Ours | **31.5** | **23.6** | **15.3** |

instance-level label-based methods relying on distillation, such as BIND (Zhang et al., 2024a). Notably, it also outperforms several recent distillation-based methods, such as CCKT-Det (Zhang et al., 2025a) and OV-DQUO (Wang et al., 2025), under the same ResNet-50 backbone. On OV-LVIS (Table 3), CoT-PL achieves strong performance (23.0% $AP_r$) using our generated pseudo-annotations under hand-crafted prompts. It significantly outperforms ViLD (Gu et al., 2022), Detic (Zhou et al., 2022), and BARON (Wu et al., 2023c), and performs better than more recent methods such as LBP (Li et al., 2024) and MIC (Wang et al., 2023c), which utilize extra data with 100 class names. These results highlight the effectiveness of our pseudo-labeling framework across diverse settings.

**Statistics.** Table 2 provides detailed statistics of pseudo-labels generated by different MLLM variants on two benchmarks. Our pseudo-labels span around 4K diverse object classes, totaling 637.349 annotations across these benchmarks. While BLIP2 (Li et al., 2023) covers the most categories (6,036), its relatively low annotation count (395K) suggests sparse predictions with lower per-class confidence. In contrast, InstructBLIP (Dai et al., 2023) produces fewer categories (3,106) but more annotations (566K), reflecting more confident labeling. Qwen2 (Bai et al., 2023) achieves the highest annotation density, generating 637K annotations across 3,916 categories. Notably, as the quality of teacher models improves, the number of "Unsure" responses from MLLMs decreases, yielding more confident pseudo-labeling. These results show that stronger teacher models with higher annotation density with fewer "Unsure" responses, producing high-quality pseudo-labels for OVD.

## 4.2 ABLATION ANALYSIS

**Quality of pseudo-labels for complex scenes.** We compare the quality of our pseudo-labels with prior state-of-the-art methods (Gao et al., 2022; Zhao et al., 2022; 2024). For a fair comparison, we adopt the same experimental setup and report $AP_{50}^N$ on the COCO validation set. As shown in Table 4, existing methods perform reasonably well on COCO Novel but degrade significantly under challenging scenarios such as *Crowded* and *Occluded*. Following (Lin et al., 2014; Qi et al., 2022), we define *Crowded* as images with more than 8 objects (COCO average), and *Occluded* as novel ground-truth (GT) boxes covered more than 50% by other GT boxes. These results suggest that previous pseudo-labeling methods lack the reasoning capabilities necessary for fine-grained visual understanding, resulting in noisy pseudo-labels. In contrast, our method achieves superior performance in such challenging scenarios through visual chain-of-thought reasoning.

**Impact of the individual proposed modules.** We conduct an ablation study on OV-COCO to assess the contribution of each component in our CoT-PL framework: pseudo-labeling with CoT reasoning and contrastive background learning (CBL). As shown in Table 5, naïvely prompting MLLMs with a single-step query for both class names and background grounding without image preprocessing results in only a marginal 0.6% gain over the baseline (Wu et al., 2023c). In contrast, applying image preprocessing improves $AP_{50}^N$ by 2.6%, while a three-step CoT ($3\times$) further increases the gain to 5.7%. Furthermore, incorporating CBL improves performance by 7.1% via better feature disentanglement between objects and background. It also yields a 2.9% gain in the single-step setting with more unlabeled objects. These findings demonstrate the effectiveness of our pseudo-label generation pipeline and CBL in generating high-quality pseudo-labels.

**Impact of semantic anchors.** We assess the impact of semantic anchors under different thresholds in Table 6. The ALL setting, which uses all pseudo-labels without filtering, results in a slight per-

Table 5: **Ablation of the proposed individual modules.** CoT (K×) refers to pseudo-labeling guided by K-step chain-of-thought (CoT) reasoning. † denotes no image preprocessing.

| CoT† (1×) | CoT (1×) | CoT (3×) | CBL | $AP_{50}^N$ (%) |
|:---:|:---:|:---:|:---:|:---:|
| ✓ | - | - | - | 34.6 |
| - | ✓ | - | - | 37.2 |
| - | ✓ | - | ✓ | 40.1 |
| - | - | ✓ | - | 40.3 |
| - | - | ✓ | ✓ | **41.7** |

Table 6: **Impact of semantic anchors.** MIN denotes the fewest base-class annotation, while ALL uses all pseudo labels. CoT-PL performs best under MIN with more reliable pseudo-labels on OV-COCO.

| Thresholds | $AP_{50}^N$ | $AP_{50}^B$ |
|:---:|:---:|:---:|
| ALL | 40.7 | 59.1 |
| MIN | 41.7 | 59.4 |

Table 7: **Impact of proposal generators on pseudo-labeling.** The detector is trained on pseudo-labels from multiple proposal generators.

| Proposal Generator | COCO Novel | LVIS |
|:---|:---:|:---:|
| Mask R-CNN (2017) | 39.7 | 21.6 |
| MAVL (2022) | 40.9 | 22.1 |
| SAM (2023) | **41.7** | **23.0** |

Table 8: Comparison of MLLM variants for pseudo-labeling. **Best** and second best results are highlighted.

| Model | Size | $AP_{50}^N$ | $AP_{50}^B$ |
|:---|:---:|:---:|:---:|
| BLIP2 (2023) | 2.7B | 37.6 | 59.5 |
| InstructBLIP (2023) | 7B | 40.1 | 58.7 |
| Qwen2 (2023) | 7B | **41.7** | 59.4 |

formance drop due to the inclusion of noisy and uncertain predictions. In contrast, the MIN setting, which selects semantic anchors with the fewest base-class annotations, achieves improved performance. These results suggest that semantic anchors help filter out unreliable and sparse predictions while guiding training toward semantically consistent regions.

**Impact of pseudo-box generator quality.** The proposed pseudo-labeling pipeline uses region proposals generated by class-agnostic SAM (Kirillov et al., 2023). Following PB-OVD (Gao et al., 2022), we compare SAM with other proposal generators, such as Mask R-CNN (He et al., 2017) and MAVL (Maaz et al., 2022), for pseudo-labeling. Table 7 shows that SAM achieves the best results on both COCO Novel (41.7%) and LVIS (23.0%), highlighting its potential as a proposal generator for pseudo-labeling. Notably, our method achieves competitive performance even with different proposal generators, demonstrating the effectiveness of structured chain-of-thought reasoning.

**Comparison of MLLM variants.** We compare different MLLM variants used in our pseudo-labeling pipeline, as shown in Table 8. BLIP-2 (Li et al., 2023), with the smallest parameter size (2.7B), yields the lowest detection performance, partially due to frequent "Unsure" responses, indicating limited capability in generating reliable pseudo-labels. In contrast, both InstructBLIP (Dai et al., 2023) and Qwen2 (Bai et al., 2023) have a comparable model size (7B), but Qwen2 consistently outperforms InstructBLIP across standard multimodal benchmarks such as MMBench (Liu et al., 2024c). This performance gap suggests that stronger MLLMs generate higher-quality pseudo-labels, leading to a 1.6% improvement in $AP_{50}^N$ for novel object detection. These findings indicate that advances in MLLM capability can translate into roughly linear gains within our framework.

## 5 CONCLUSION

We introduce CoT-PL, a new pseudo-labeling framework for open-vocabulary object detection (OVD) that leverages structured visual chain-of-thought (CoT) reasoning. Harnessing the zero-shot reasoning capabilities of multimodal large language models (MLLMs), CoT-PL decomposes object understanding into three interpretable reasoning steps: (1) object recognition, (2) caption-free zero-shot labeling, and (3) background extraction. The third step further leads to our contrastive background learning (CBL) that mitigates background collapse by disentangling object and background features. As a result, CoT-PL, a unified system integrating CoT reasoning with CBL, generates high-quality pseudo-labels in complex visual scenes and consistently improves performance with stronger teacher MLLMs. CoT-PL achieves state-of-the-art results across multiple OVD benchmarks. We hope our work inspires further exploration of visual CoT reasoning in downstream perception tasks.

## REPRODUCIBILITY STATEMENT

All materials required to reproduce our results are documented in the main paper and the appendices. Specifically, the skeleton code and scripts are described in Appendix A. To preserve anonymity, the skeleton code only illustrates the methodological flow. The workable version will be provided upon publication. The device specifications are provided in Appendix C, and the implementation details, including the hyperparameters, schedules, and optimizer settings, are presented in Section 4 and Appendix E. In addition, the dataset information is provided in Appendix F, the prompts used in the CoT pipeline are specified in Appendix J, and the pre- and post-processing setup is detailed in Appendix H.1. Finally, the pseudo-label statistics are reported in Table 2 and Appendix H.1.

## USAGE OF LARGE LANGUAGE MODELS

We utilize large language models, such as ChatGPT (Hurst et al., 2024), to aid or polish writing only. This process primarily includes spell-check and paraphrasing a few texts for clarity and brevity. We do not use any pre-defined prompt to query the model. Rather, we simply ask from scratch with a brief prompt (*i.e.,* `{input_sentence}` `Question:` `Please polish this sentence` `concisely.`). All outputs are thoroughly human-reviewed before inclusion in the paper. All co-authors participate in this review process.

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

# APPENDIX

## A    CODE & REPRODUCTION

To preserve anonymity, the full codebase is provided as a ZIP file in the supplementary materials on the submission site. Instructions for running the code are included in the README.md file within the ZIP archive. In accordance with code submission requirements, which include anonymity and no web pointers, the provided skeleton code is intended solely to demonstrate the methodological flow. The workable version will be provided upon publication.

## B    LIMITATIONS & FUTURE WORK

Despite its effectiveness, CoT-PL faces two limitation. (1) It depends on the capabilities of a sufficiently strong MLLM; less capable models produce lower-quality pseudo-labels and degrade detection performance. To mitigate this, it treats the MLLM's three-way judgment (Yes/No/Unsure) as a hard gate, safely discarding all potentially false regions labeled Unsure. This rigid design risks accumulating irreversible false negatives. (2) Long-tailed categories are unfairly removed by the frequency threshold of the minimum number of annotations in base classes, discarding any labels below this count as noise for simplicity. Exploring these issues serves as a promising direction for future work.

## C    DEVICE INFORMATION

All experiments were conducted using eight NVIDIA A6000 GPUs with PyTorch 1.12.1. Each training run took approximately 11 hours and involved around 12K GPU memory usage. For a fair comparison with the baseline, we fixed the random seed to 1194806617 across all experiments to ensure reproducibility.

## D    IMPLEMENTATION DETAILS

**Baseline.**    We build our CoT-PL upon Faster R-CNN (Ren et al., 2015) with a ResNet-50 FPN backbone, consistent with prior open-vocabulary object detection (OVD) work. The backbone is initialized using weights pre-trained with SOCO (Wei et al., 2021) and utilizes synchronized Batch Normalization (SyncBN) (Zhang et al., 2018). We adopt the $1\times$ training schedule for OV-COCO (Lin et al., 2014) and $2\times$ for OV-LVIS (Gupta et al., 2019).

**Pseudo-label generation process.**    During our offline pseudo-label generation process, we leverage SAM (Kirillov et al., 2023; Qin et al., 2024) to produce class-agnostic object proposals and apply Qwen2 (7B) (Bai et al., 2023) as our default MLLM for visual chain-of-thought (CoT) reasoning. CLIP based on ViT-B/16 (Radford et al., 2021) is used to encode textual prompts, which are constructed using the hand-crafted template "`a photo of [OBJ]`" following ViLD (Gu et al., 2022). Pseudo-labels are generated using only the training set, without leveraging any image captions. The pseudo-labels are used exclusively during training and discarded during inference.

**Contrastive Background Learning.**    Following ViLD, the contrastive background learning (CBL) background prototypes are built from hand-crafted prompts rather than category names. Using prompt engineering, we define five generic background types, such as sky, water surface, vegetation, paved ground, and plain wall, and craft a small set of object-free prompts for each (*i.e.* "clear sky background, no objects"). We tokenize the prompts and encode them using the CLIP text encoder, then average the resulting text embeddings to obtain a single prototype per background type.

## E    HYPERPARAMETERS

For fair comparison, we adopt the same hyperparameter settings as BARON. We use the SGD optimizer with a momentum of 0.9 and a weight decay of $2.5 \times 10^{-5}$. The initial learning rate is set to 0.04 for OV-COCO and 0.08 for OV-LVIS. Models are trained for 90,000 iterations on OV-COCO (Lin et al., 2014) and 180,000 iterations on OV-LVIS (Gupta et al., 2019), with a fixed batch

size of 16 across all experiments. During training, model checkpoints are saved every 10,000 iterations for OV-COCO and every 30,000 iterations for OV-LVIS. The best-performing checkpoint on the validation set is selected for final evaluation.

For our proposed modules, we provide the hyperparameter configurations used in the OV-COCO (Lin et al., 2014) and OV-LVIS (Gupta et al., 2019) experiments. For semantic anchor construction, we filter out infrequent pseudo-labels using a minimum annotation threshold—set to 1,237 for OV-COCO and 1 for OV-LVIS. Additionally, the background contrastive loss temperature parameter is set to $\tau'' = 5.0$, which controls the regularization strength of background embeddings relative to foreground embeddings.

## F  DATA APPENDIX

We report the statistics of the benchmarks used in this study. For both OVD benchmarks, our pseudo-label generation pipeline uses only the training set, while the validation and test sets remain unused to ensure fair evaluation during inference.

- **OV-COCO** (Lin et al., 2014): OV-COCO is based on the COCO 2017 detection split, containing 118,287 training and 5,000 validation images with instance-level bounding box annotations. We follow the category split from OVR-CNN (Zareian et al., 2021), which defines 48 base and 17 novel categories. During pseudo-label generation, images containing only novel categories are discarded to ensure consistent supervision. All pseudo-labels are generated solely from the base training split.

- **OV-LVIS** (Gupta et al., 2019): OV-LVIS is derived from LVIS v1.0, comprising over 120K training images with annotations for 1,203 categories. Following the ViLD (Gu et al., 2022) protocol, we treat 337 rare categories as novel, and the remaining frequent and common categories as base. Due to the severe long-tail distribution, some rare categories contain fewer than five instances; such categories are removed during semantic anchor construction. Evaluation is performed on the standard LVIS validation split.

- **Objects365** (Shao et al., 2019a): Objects365 contains 1,742,289 training and 80,000 validation images across 365 object categories. We use this dataset solely for cross-dataset evaluation without any additional fine-tuning. Specifically, the model trained on OV-LVIS is directly evaluated on the Objects365 validation split to assess its transferability. All category names are mapped via exact string matching using CLIP prompt templates.

## G  BASELINES

**Open-vocabulary detectors.** Recent advances (Ren et al., 2015; He et al., 2017) in open-vocabulary object detection (OVD) have been largely driven by the emergence of foundation models, including vision-language models (VLMs) (Radford et al., 2021; Jia et al., 2021). VLMs support novel class recognition in OVD through various techniques, such as pseudo-labeling. We build upon Faster R-CNN (Ren et al., 2015) for OVD, replacing its classifier with a linear layer that projects region features into the word embedding space. This enables each region to be represented by multiple pseudo-words, capturing the rich semantics of each object. Given $C$ object categories, the probability of a region being classified as the $c$-th category:

$$p_c = \frac{\exp(\tau \cdot \langle \mathcal{T}(w), f_c \rangle)}{\sum_{i=0}^{C-1} \exp(\tau \cdot \langle \mathcal{T}(w), f_i \rangle)}, \tag{4}$$

where $\mathcal{T}$ is the text encoder, $\langle \cdot, \cdot \rangle$ denotes cosine similarity, $\tau$ is a temperature scaling factor, $\mathcal{T}(w)$ represents the text embedding of pseudo-words, and $f_c$ is the category embedding of a prompt template encoded by the text encoder (*i.e.* "a photo of {category} in the scene").

**BARON (Wu et al., 2023c).** Additionally, we instantiate the idea of BARON (Wu et al., 2023c) to capture compositional structures for simplicity. During training, it learns using Faster R-CNN's regression and classification losses (Ren et al., 2015), with annotations provided only for the base set. BARON first groups contextually related neighboring regions for each region proposal extracted from Faster R-CNN (Ren et al., 2015), forming a *bag of regions*. BARON then projects these

**(a) Aligning bag of regions**

**(a) Aligning bag of regions (with semantic anchors)**

Figure 5: **The overall architecture of BARON (Wu et al., 2023c)**. To reduce sampling time during training, BARON's naive neighbor sampling can optionally be replaced with our semantic anchor-based strategy. In particular, caching the anchors reduced training time by 25% compared to the baseline, while maintaining the original performance.

regions into the word embedding space using the linear layer within Faster R-CNN, resulting in pseudo-words. The pseudo-words are passed through the text encoder to obtain the bag-of-regions embedding $f_t^i = \mathcal{T}(w_0^i + p_0^i, w_1^i + p_1^i, \cdots, w_{N^i-1}^i + p_{N^i-1}^i)$, where $N^i$ is the number of regions in the $i$-th bag, $p_j^i$ represents the positional embedding of the $j$-th region in the $i$-th bag. Finally, this bag-of-regions embedding is aligned with the VLM's image embeddings $f_v^i = \mathcal{V}(b_0^i, b_1^i, \cdots, b_{N_i}^i)$, where $b_j^i$ is the $j$-th region in the $i$-th bag. To align the bag-of-regions embeddings, BARON employs a contrastive learning loss based on InfoNCE (Rusak et al., 2024):

$$\mathcal{L}_{bag} = -\frac{1}{2} \sum_{k=0}^{G-1} \big( \log(p_{t,v}^k) + \log(p_{v,t}^k) \big), \tag{5}$$

$$p_{t,v}^k = \frac{\exp(\tau' \cdot \langle f_t^k, f_v^k \rangle)}{\sum_{i=0}^{G-1} \exp(\tau' \cdot \langle f_t^k, f_v^i \rangle)}, \tag{6}$$

$$p_{v,t}^k = \frac{\exp(\tau' \cdot \langle f_v^k, f_t^k \rangle)}{\sum_{i=0}^{G-1} \exp(\tau' \cdot \langle f_v^k, f_t^i \rangle)}, \tag{7}$$

where $G$ is the number of bags for each region proposal and $\tau'$ is a temperature scaling factor. This allows BARON to leverage the compositional structures inherent in VLMs, resulting in moderate performance gains.

To align individual region embeddings, BARON uses a contrastive learning loss similar to InfoNCE:

$$\mathcal{L}_{\text{individual}} = -\frac{1}{2} \sum_{k=0}^{N-1} \big( \log(q_{t,v}^k) + \log(q_{v,t}^k) \big), \tag{8}$$

$$q_{t,v}^k = \frac{\exp(\tau_{\text{individual}} \cdot \langle g_t^k, g_v^k \rangle)}{\sum_{i=0}^{N-1} \exp(\tau_{\text{individual}} \cdot \langle g_t^k, g_v^i \rangle)}, \tag{9}$$

$$q_{v,t}^k = \frac{\exp(\tau_{\text{individual}} \cdot \langle g_v^k, g_t^k \rangle)}{\sum_{i=0}^{N-1} \exp(\tau_{\text{individual}} \cdot \langle g_v^k, g_t^i \rangle)}, \tag{10}$$

where $N$ is the total number of regions $g_t^k$ and $g_v^k$ are the teacher and student embeddings for the $k$-th region, and $\tau_{\text{individual}}$ is the temperature parameter to re-scale the cosine similarity.

**Segment Anything Model (SAM).** SAM (Kirillov et al., 2023) is a general-purpose segmentation model that predicts instance masks given spatial prompts. It enables high-quality, class-agnostic mask generation via zero-shot segmentation, providing fine-grained object candidates valuable for downstream tasks (Yuan et al., 2024; Han et al., 2025). Recently, LangSplat (Qin et al., 2024) leveraged SAM to extract hierarchical segmentation masks from images, enabling structured multi-scale object representation. By densely sampling point prompts across the image, SAM generates

Table 9: **Zero-shot performance of MLLMs across academic benchmarks.** Average accuracy across six multimodal evaluation datasets: MMBench v1.1 (Test$_{CN}$/Test$_{EN}$), MMStar, MMMU (Val), and HallusionBench.

| Model | MMBench V1.1 (2024c) | MMStar (2024) | MMMU (2024) | HallusionBench Avg. (2024) |
|---|---|---|---|---|
| BLIP2 (2.7B) (2023) | - | - | - | - |
| InstructBLIP-7B (2023) | 28.4 | 32.7 | 30.6 | 31.2 |
| Qwen2-VL-7B (2023) | 81.0 | 60.7 | 53.7 | 50.4 |

a diverse set of masks that capture object regions at varying levels of granularity. These masks are filtered and organized into three semantic levels—subpart, part, and whole—based on confidence and spatial criteria. This hierarchical masking facilitates precise, pixel-aligned feature extraction and supports the construction of language-aware 3D scene representations.

**Multimodel Large Language Models.** Multimodal large language models (MLLMs) have recently been explored in this context (Zhou et al., 2025; Zhang et al., 2025a; Yin et al., 2023; Wang et al., 2023a; Wu et al., 2023b), as they integrate visual perception with language-based reasoning and instruction following.

- **BLIP-2:** BLIP-2 (Li et al., 2023) adopts a modular architecture comprising a frozen image encoder, a trainable QFormer (Zhang et al., 2024b), and a frozen language model such as OPT (Zhang et al., 2022). This setup enables efficient vision-language alignment and achieves strong performance on tasks such as image captioning and visual question answering (VQA) with minimal training.

- **InstructBLIP:** InstructBLIP (Dai et al., 2023) builds on BLIP-2 (Li et al., 2023) via instruction tuning, integrating a ViT-G vision encoder (Dosovitskiy et al., 2021) and a frozen language model, such as Flan-T5 (Chung et al., 2024). This design allows the model to follow natural language instructions and generalize across diverse multimodal tasks. As shown in Table 9, this MLLM exhibits fair zero-shot performance on academic multimodal benchmarks, with accuracy ranging from 24% to 32% on most tasks.

- **Qwen2:** Qwen2 (Bai et al., 2023) is a multilingual large language model series ranging from 0.5B to 72B parameters, trained on diverse and high-quality web-scale data. It employs a tokenizer optimized for multilingual understanding and exhibits strong performance in reasoning, instruction following, and general language tasks. As shown in Table 9, the model demonstrates strong generalization, particularly on MMBench (Liu et al., 2024c) (81.0%), indicating superior reasoning capabilities and vision-language alignment.

## H ADDITIONAL ABLATION STUDY

### H.1 IMAGE PREPROCESSING

We explore image preprocessing strategies to help MLLMs better focus on target regions. Their impact on detection performance is summarized in Table 10.

MLLMs (Li et al., 2023; Dai et al., 2023; Bai et al., 2023; Wang et al., 2023a) exhibit strong zero-shot reasoning across vision-language tasks such as image captioning and retrieval. However, when applied to object-level understanding, recent studies (Zang et al., 2025; Fu et al., 2024) have shown that they remain highly sensitive to visual context. In our setting, where the MLLM is prompted on individual region proposals, it is essential to emphasize the target object while suppressing irrelevant background information. As shown in Table 10, omitting preprocessing slightly degrades performance, yielding an $AP_{50}^{N}$ of 33.6 relative to the baseline.

A naïve solution is to black out pixels outside the target segmentation mask. While this approach directs the model's attention to the target region, it removes surrounding context and may cause misclassification due to silhouette artifacts—a limitation highlighted in prior work (Chang et al., 2023; Fontanini et al., 2023). For example, as illustrated in Figure 6-b, the model misclassifies a tree as a giraffe, influenced by the masked silhouette of a giraffe in the background. Despite this issue, the black mask strategy significantly improves performance, achieving an $AP_{50}^{N}$ of 38.5 (+4.5).

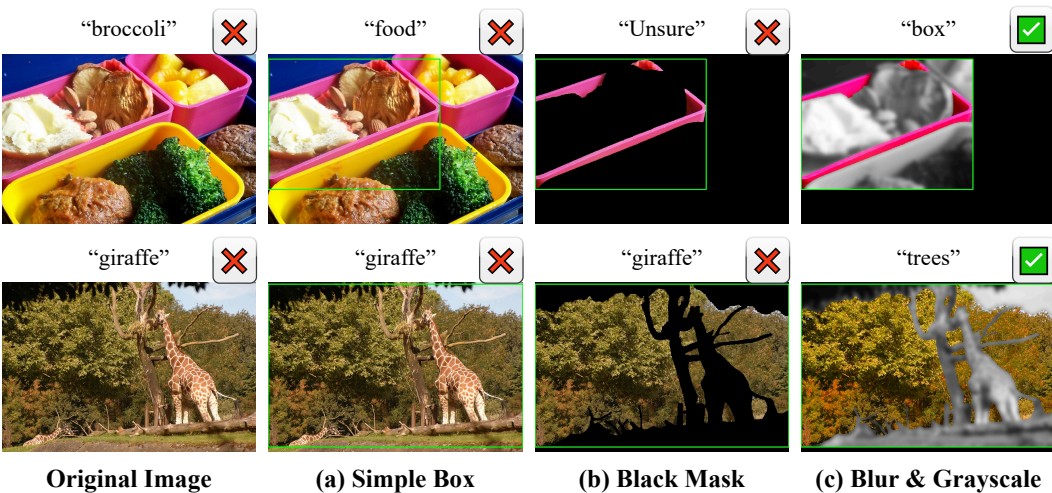

Figure 6: **Visualization of image preprocessing strategies.** We adopt three strategies: **(a)** simple box, **(b)** black mask, and **(c)** blur & grayscale. Each image is labeled with the MLLM's (Bai et al., 2023) prediction. Qualitative analysis indicates that (c) yields the most reliable zero-shot object recognition performance.

Table 10: **Effect of image preprocessing on pseudo-label quality.** Blurred and grayscale images improve MLLM reasoning, leading to the best results on OV-COCO.

| Method | $\text{AP}_{50}^{N}$ (%) | $\text{AP}_{50}^{B}$ (%) |
|---|---|---|
| BARON (Wu et al., 2023c) | 34.0 | 60.4 |
| Simple Box | 33.6 (-0.4) | 59.6 |
| + Masked image | 38.5 (+4.5) | 59.5 |
| + Blurred & grayscale | **41.7** (+7.7) | 59.4 |

To mitigate such misclassifications, prior work (Qin et al., 2024) suggests that grayscaling and blurring regions outside the mask can effectively suppress background noise and enhance model focus. In practice, we validate that this strategy improves localization and reasoning in MLLMs (Bai et al., 2023), as shown in Table 10. Building on these findings, we adopt this preprocessing approach for all region proposals queried by the MLLM in our CoT pipeline. This configuration yields the best performance among all settings, achieving an $\text{AP}_{50}^{N}$ of 41.7 (+7.7).

- **Raw image:** Original training images from the OVD benchmark, where each proposal box generated by SAM is preserved and all pixels outside the box are blacked out. We observe that MLLMs often struggle to focus on the proposal region during reasoning over the input query.

- **Masked image:** Starting from the raw image, only the segmentation mask region within each proposal box is retained, while all other pixels inside the box are masked in black. This often distracts the model due to black silhouettes (e.g., a masked giraffe shape), rather than helping it focus on the intended region.

- **Blurred and grayscale image:** Also based on the raw image, the segmentation mask region within each proposal box is preserved, while surrounding pixels inside the box are blurred and converted to grayscale. Pixels outside the box are blacked out. We adopt this preprocessing technique in our pipeline, as it effectively preserves contextual cues while maintaining focus on the target region. For reproducibility, we apply standard BGR-to-grayscale conversion and Gaussian blur with a kernel size of 31×31 and a sigma of 0.

**Statistics.** Figure 7 illustrates the annotation counts of pseudo-labels in the OV-COCO dataset, revealing a typical long-tail distribution. A small number of frequent categories account for the

Table 11: **Statistics of pseudo-labels per super-class.** Super-classes are derived from pseudo-labels using GPT-4o (Hurst et al., 2024) and Qwen2 (Bai et al., 2023) on OV-COCO.

| Animals | Furniture | Weapons/Tools | Vehicles | Electronics | Food/Drink |
|---------|-----------|---------------|----------|-------------|------------|
| 11 | 14 | 5 | 6 | 6 | 6 |
| **Buildings** | **Clothing** | **Shapes** | **Sports** | **Misc.** | |
| 3 | 3 | 3 | 3 | 5 | |

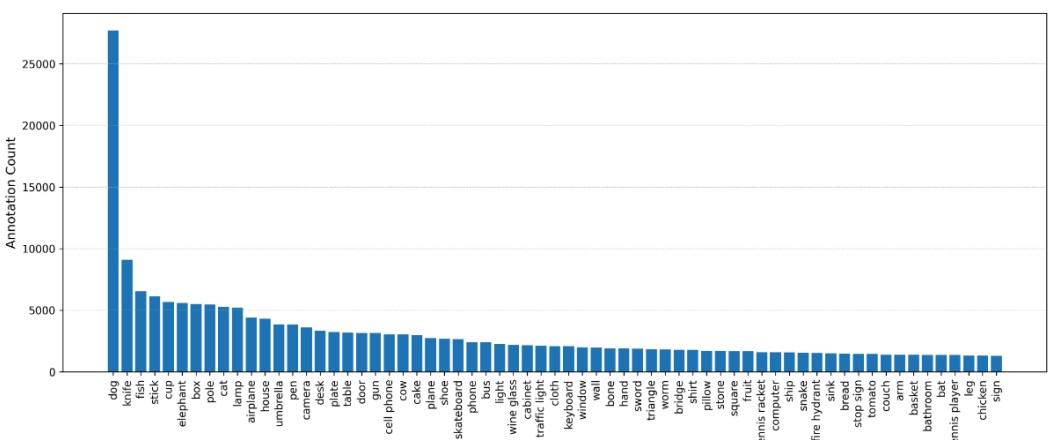

Figure 7: **Distribution of annotations per class.** Based on our Qwen2 (Bai et al., 2023) pseudo-labels across 65 classes in the OV-COCO benchmark. For brevity, we omit the OV-LVIS distribution visualization, as it includes over 3,000 different pseudo-labels.

majority of annotations, reflecting their higher prevalence in the training data. This imbalance naturally emerges, as the MLLM tends to predict commonly occurring and semantically salient objects. Notably, the pseudo-labels also include several novel categories (*e.g.*, "dog," "knife," and "cup"), which contribute to improved detection performance on these previously unseen classes.

**Semantic diversity of pseudo-labels.**  To better understand the nature of our pseudo-labels, we further group them into several semantic super-classes using GPT-4o (Hurst et al., 2024), prompted with: "Question: Group the classes into broader super-classes." As shown in Table 11, prominent groups such as *Furniture* and *Animals* emerge, highlighting the robustness of our pipeline in predicting semantically diverse object categories.

However, we observe that some categories—particularly abstract or non-object-level concepts such as *Shapes* and *Misc*—often result in vague or inconsistent outputs. This suggests that current MLLMs are not yet fully equipped to handle such concepts reliably, indicating potential for future improvement

### H.2    TRANSFER DETECTION PERFORMANCE

To evaluate cross-dataset generalization, we follow the transfer detection setting, where the model is trained on OV-LVIS and evaluated on COCO (Lin et al., 2014) and Objects365 (Shao et al., 2019b) without any additional fine-tuning. As shown in Table 12, our CoT-PL, using the default ResNet-50 backbone, demonstrates strong transferability—surpassing F-VLM (Kuo et al., 2022) by +4.3% and +2.0% AP on COCO and Objects365, respectively, and outperforming BARON (Wu et al., 2023c) by +0.6% and +0.3%. Notably, CoT-PL also substantially reduces the performance gap with fully supervised detectors, narrowing the AP difference to only 9.7% on COCO and 11.7% on Objects365.

Table 12: **Result comparisons of the LVIS-trained model on COCO (Lin et al., 2014) and Objects365 (Shao et al., 2019b).** We use BARON as the baseline and evaluate all methods without fine-tuning.

| Methods | MS-COCO (Lin et al., 2014) | | | Objects365 (Shao et al., 2019b) | | |
|---|---|---|---|---|---|---|
| | AP (%) | AP$_{50}$ (%) | AP$_{75}$ (%) | AP (%) | AP$_{50}$ (%) | AP$_{75}$ (%) |
| Supervised (Gu et al., 2022) | 46.5 | 67.6 | 50.9 | 25.6 | 38.6 | 28.0 |
| ViLD (Gu et al., 2022) | 36.6 | 55.6 | **39.8** | 11.8 | 18.2 | 12.6 |
| DetPro (Du et al., 2022) | 34.9 | 53.8 | 37.4 | 12.1 | 18.8 | 12.9 |
| F-VLM (Kuo et al., 2022) | 32.5 | 53.1 | 34.6 | 11.9 | 19.2 | 12.6 |
| BARON (Wu et al., 2023c) | 36.2 | 55.7 | 39.1 | 13.6 | 21.0 | 14.5 |
| CoT-PL (Ours) | **36.8** | **56.0** | 39.5 | **13.9** | **21.6** | **14.9** |

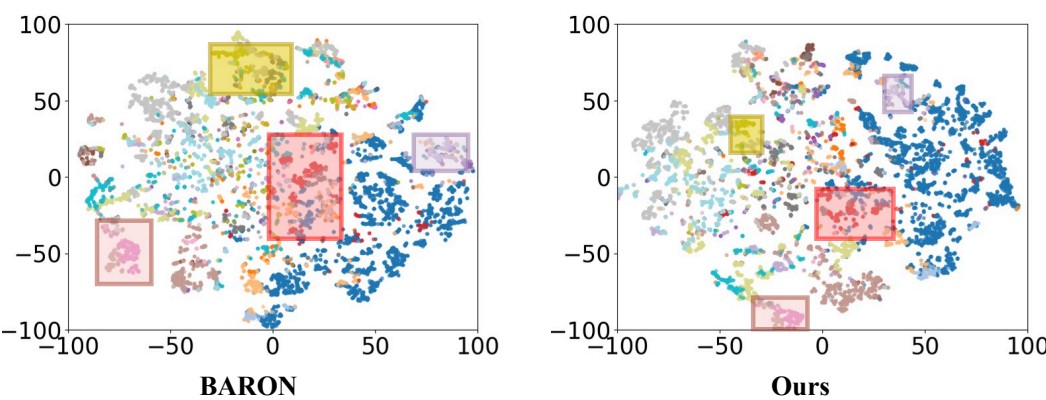

**BARON**    **Ours**

Figure 8: **t-SNE results of feature distributions.** Compared to BARON (Wu et al., 2023c), our CoT-PL generates more compact embeddings for novel representations.

### H.3 t-SNE Visualization

We employ t-SNE (van der Maaten & Hinton, 2008) to visualize the feature distribution of novel category proposals, emphasizing the effectiveness of our designed schemes. Figure 8 demonstrates that CoT-PL enables the detector to learn more compact and discriminative features for novel category proposals than BARON (Wu et al., 2023c). Gray points represent base classes, while novel classes are shown in color.

### H.4 Background Feature Disentanglement

We employ t-SNE (van der Maaten & Hinton, 2008) to visualize the feature distribution of novel category proposals and background regions in Figure 9. Compared to BARON, CoT-PL more effectively separates novel class objects from the background. Pink points indicate learnable background embeddings labeled as "`__Background__`", while green points represent the novel class *airplane*.

## I Additional Qualitative Results

### I.1 Grad-CAM Visualization

In this paper, we introduce **CoT-PL**, a novel framework that integrates structured visual chain-of-thought (CoT) reasoning into the pseudo-labeling process. CoT-PL decomposes object understanding into a sequence of interpretable steps—including region perception, category recognition, and background grounding, effectively generating high-quality pseudo-labels even in complex visual scenes.

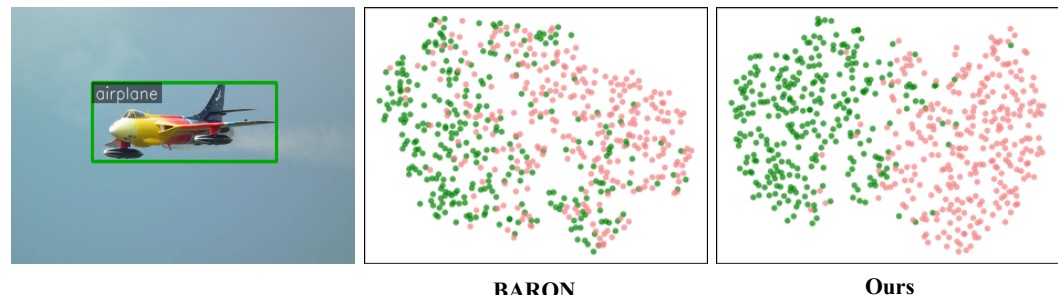

**BARON**                                    **Ours**

Figure 9: **t-SNE visualization of background separation.** The novel object class "airplane" is shown in green, and the "␣␣Background␣␣" in pink. Compared to BARON (Wu et al., 2023c), CoT-PL more effectively separates the novel class from the background.

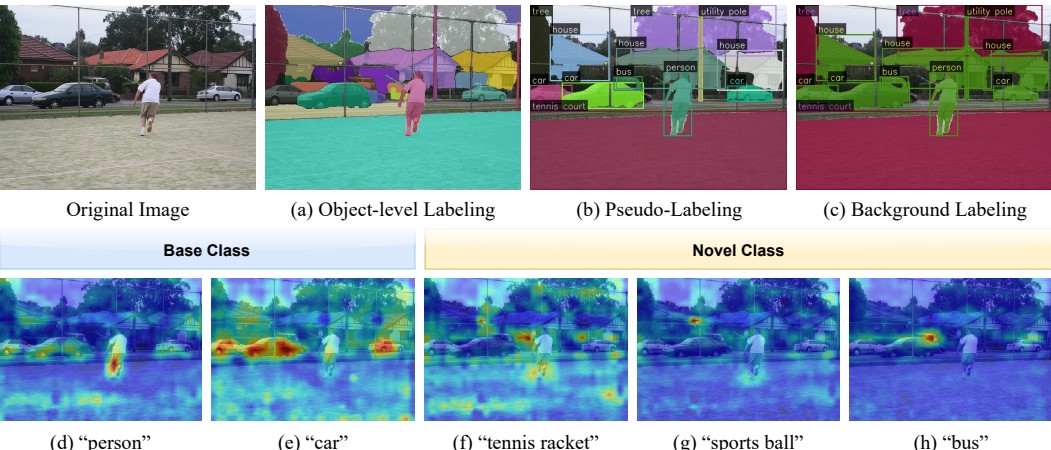

Figure 10: Our proposed CoT-PL generates accurate pseudo-labels without captions through a CoT-based MLLM pipeline: (a) verifying SAM-generated boxes as valid objects, (b) assigning zero-shot pseudo-labels, and (c) grounding boxes to distinguish objects from background. This enables detection of both base (d–e) and novel (h) classes, including unlabeled ones (f) and (g).

Unlike prior approaches, our method generates pseudo-labels solely from the training set without relying on image captions. We further propose contrastive background learning (CBL), which leverages background regions as negative samples to enhance feature disentanglement between foreground objects and background clutter.

As illustrated in Figure 10, our CoT-based MLLM pipeline proceeds by (a) verifying SAM-generated proposals as valid objects, (b) assigning zero-shot pseudo-labels, and (c) grounding each box to distinguish objects from the background. This enables our framework to detect both base and novel classes, including unlabeled categories that appear in challenging or open-vocabulary settings.

## I.2 PSEUDO-ANNOTATION VISUALIZATION

We present visualization examples of our pseudo-annotations in Figure 11. The images are sampled from the validation sets of the OVD benchmarks. Each example highlights the predicted region and corresponding class label generated by our pipeline. These visualizations demonstrate that our method produces semantically meaningful and spatially accurate pseudo-labels across diverse object categories, including both base (common) and novel (rare) classes.

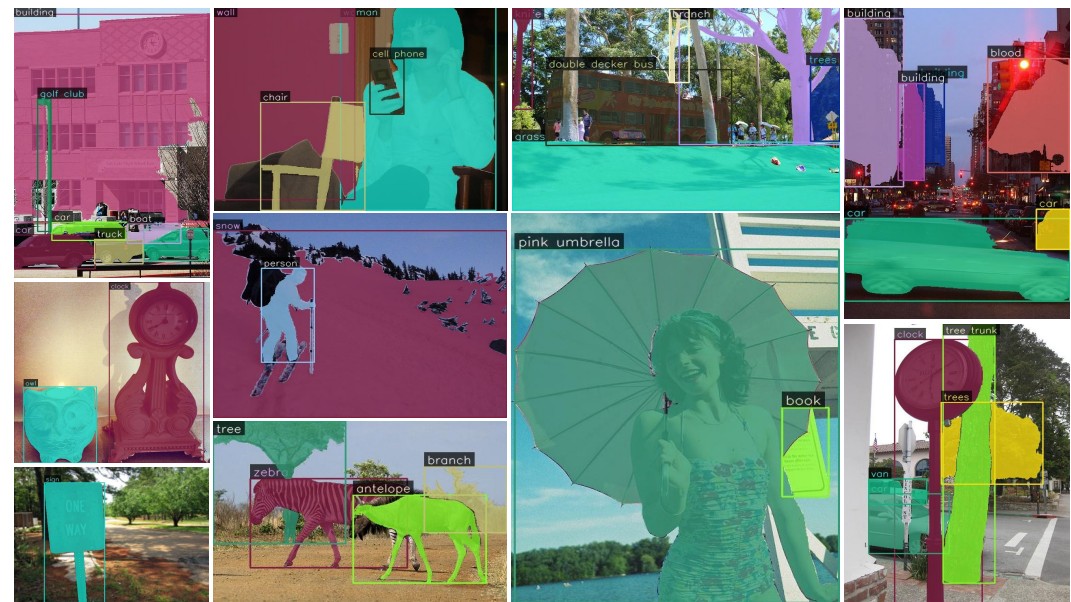

Figure 11: **Visualization of our pseudo-annotations on the OV-COCO dataset.**

### I.3 DETECTION VISUALIZATION

We present additional detection results from our method, CoT-PL, on two OVD benchmarks: OV-COCO (Lin et al., 2014) and OV-LVIS (Gupta et al., 2019), as shown in Figures 13 and 12. The images are sampled from the validation sets of each benchmark.

On COCO, CoT-PL successfully detects novel categories such as "traffic light", "bus", "keyboard", "cup", "snowboard", and "cow". On LVIS, it identifies rare categories including "boom_microphone", "mammoth", "kitchen_table", "poncho", "escargot", "shepherd_dog", and "pennant". These results demonstrate the model's ability to recognize a wide range of novel objects across both benchmarks.

## J PROMPTS

**First CoT step.** We leverage SAM's strong generalization to generate object-level pseudo boxes, which are verified by the MLLM for object presence.

```
1   Your Role: Object Presence Recognizer
2
3   You are a model that checks whether a clearly visible object exists in
        an image.
4
5   [Your task]
6   - Look at the image.
7   - If there is at least one clearly visible object, answer: Yes
8   - If there is no visible object at all (only blurred or grayscale
        areas), answer: No
9   - If it's hard to tell whether something is visible or not, answer:
        Unsure
10  - Specially, if your answer is Unsure, provide reasoning
11
12  [Important rules]
13  - Ignore blurred or grayscale areas in the image.
14  - Only consider clear, colorful, or sharply defined objects.
15
16  Your response must be only one word: Yes, No, or Unsure.
17
18  [Examples]
```

```
19   Example 1:
20   Image: (A color photo of a dog standing clearly in focus)
21   Answer: Yes
22   Reasoning: None
23
24   Example 2:
25   Image: (A grayscale image with blurred outlines and no clear shapes)
26   Answer: No
27   Reasoning: None
28
29   Example 3:
30   Image: (An image where a part of an object might be present, but it is
            not fully visible or too unclear)
31   Answer: Unsure
32   Reasoning: <YOUR_REASONING>
33
34   Now, analyze the following image:
35   Image: <attach image here>
36   Answer:
37   Reasoning:
```

**Second CoT step.** We leverage the MLLM's multi-class recognition ability to generate pseudo labels for specific concepts within each box.

```
1    Your Role: Object Category Identifier
2
3    You are a model that identifies the most likely object category that
         is clearly visible in an image.
4
5    [Your task]
6    - Look at the image.
7    - Focus only on areas that are clear, colorful, and sharply defined.
8    - Completely ignore grayscale or blurred areas.
9    - Always guess the most likely object category that is clearly visible
         .
10
11   [Instructions]
12   - Answer with only one or two words.
13   - Do not describe scenes -- just the object category.
14   - If uncertain, make your best guess based on visible clues.
15
16   [Examples]
17   Example 1:
18   Image: (A focused image of a person riding a skateboard)
19   Answer: Skateboard
20
21   Example 2:
22   Image: (A clear image of a zebra walking in grass)
23   Answer: Zebra
24
25   Example 3:
26   Image: (Blurry background, but a sharp image of a backpack is visible)
27   Answer: Backpack
28
29   Now analyze the following image:
30   Image: <attach image here>
31   Answer:
```

**Third CoT step.** We employ contrastive learning with background representations derived from a multimodal large language model (MLLM) as negative samples, encouraging the model to better separate object regions from true background areas. To obtain these background representations, we prompt the MLLM with the following instruction:

```
1    Your Role: Foreground-Background Distinguisher
2
3    You are a model that determines whether an object in an image is part
         of the foreground or the background.
```

```
4
5   [Your task]
6   - You are given an object name: "<Response>"
7   - Look at the image and decide if this object is in the foreground or
        background.
8   - Ignore any grayscale or blurred areas in the image.
9   - Use visual focus and typical object roles to decide.
10
11  [Definitions]
12  - Foreground = clearly focused subjects like people, animals, vehicles
        , or objects of interest.
13  - Background = things like sky, grass, trees, mountains, or flowers.
14
15  Your answer must be exactly one word: Foreground or Background.
16
17  [Examples]
18  Example 1:
19  Object: Dog
20  Image: (A dog is standing in sharp focus in front of a blurry park)
21  Answer: Foreground
22
23  Example 2:
24  Object: Sky
25  Image: (A person is standing in front of a bright blue sky)
26  Answer: Background
27
28  Example 3:
29  Object: Tree
30  Image: (A clear person in front, with trees in the back)
31  Answer: Background
32
33  Now analyze the following image:
34  Object: <Response>
35  Image: <attach image here>
36  Answer:
```

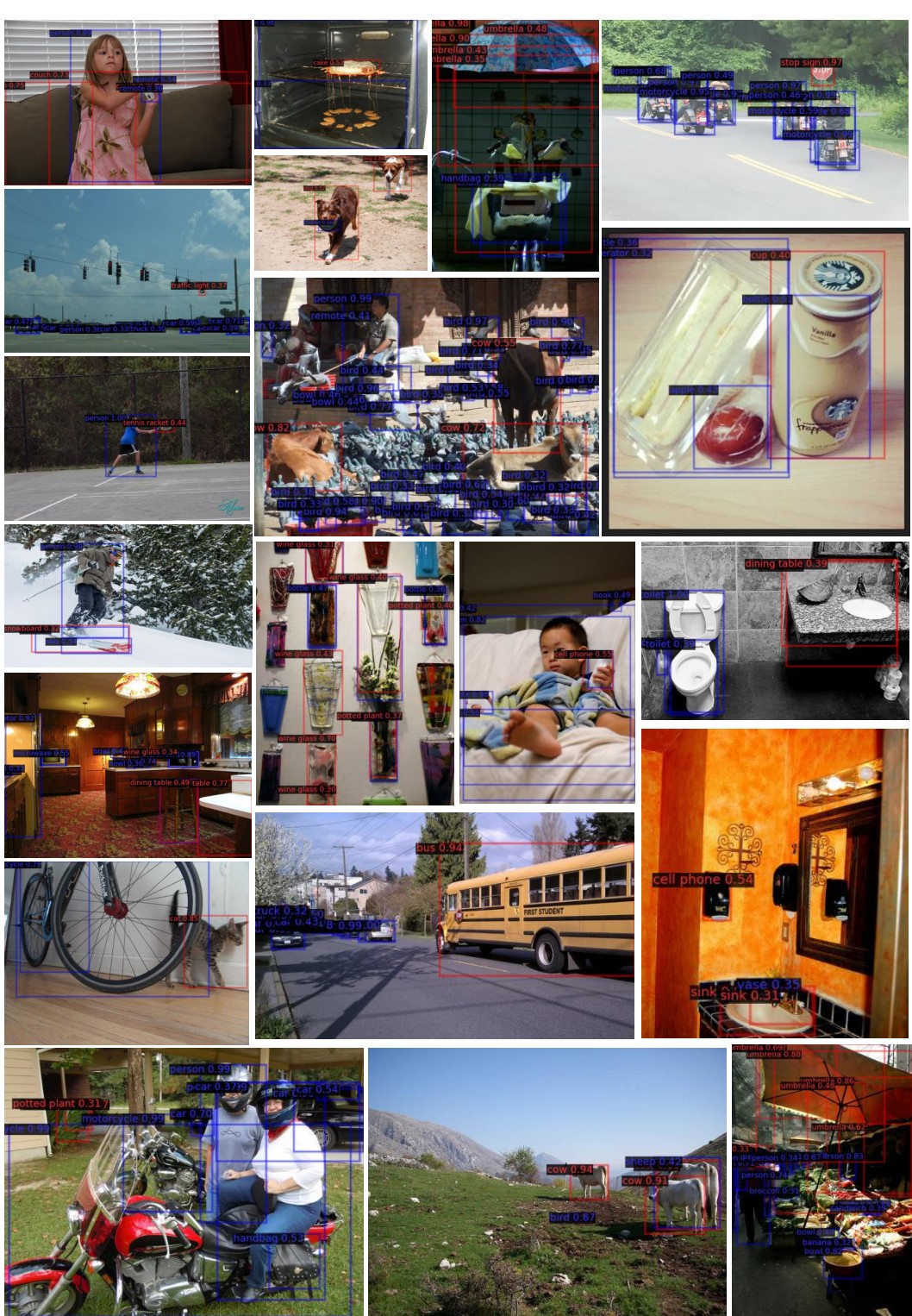

Figure 12: **Visualization of detection results on the OV-COCO dataset**. Red boxes and masks represent novel categories, while blue boxes and masks represent base categories.

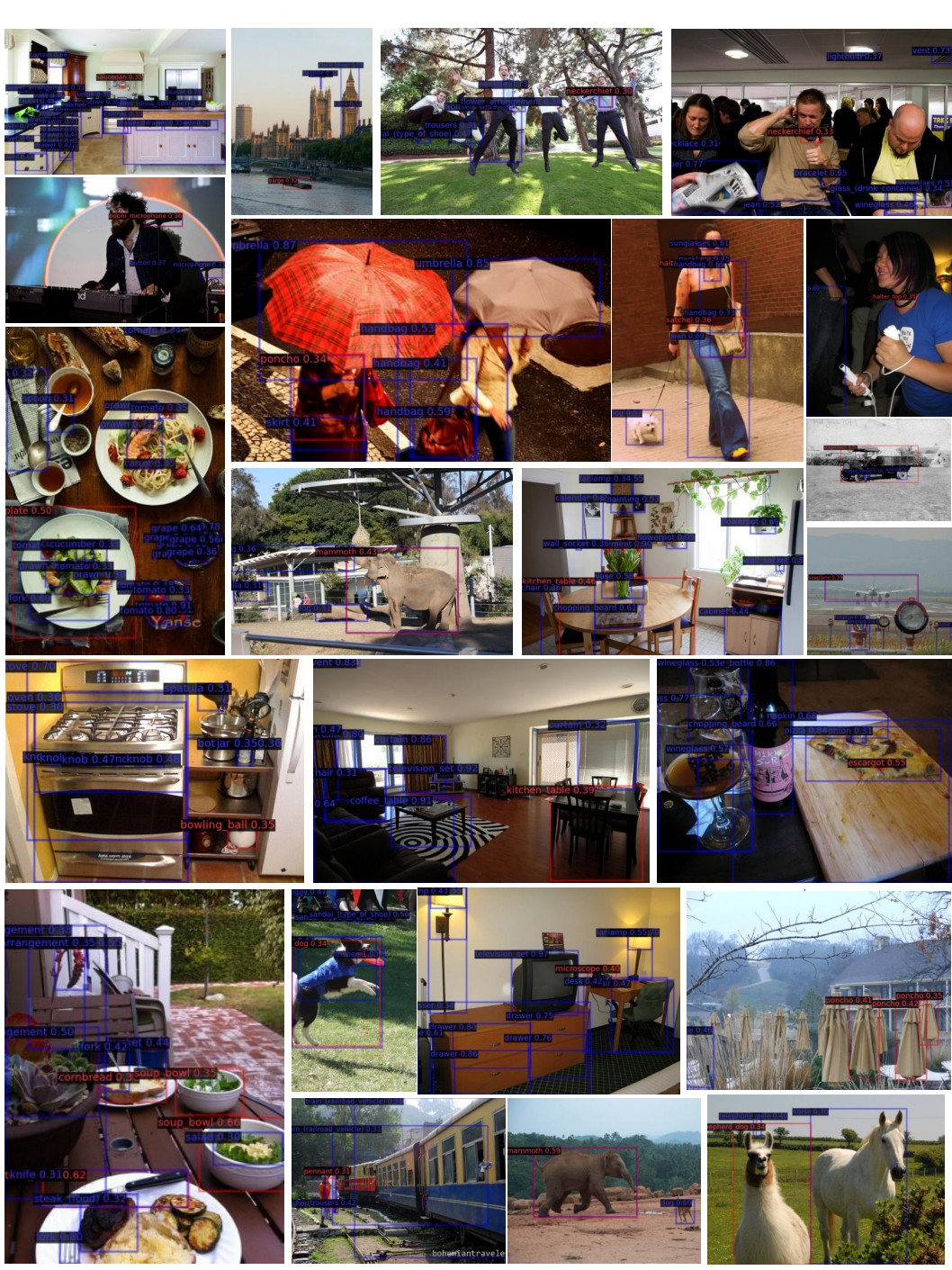

Figure 13: **Visualization of detection results on the OV-LVIS dataset**. Red boxes and masks represent novel (rare) categories, while blue boxes and masks represent base categories.

