# OpenReview forum: "CoT-PL: Visual Chain-of-Thought Reasoning Meets Pseudo-Labeling for Open-Vocabulary Object Detection"
_ICLR.cc/2026/Conference — Submitted to ICLR 2026_

### Official Review · Reviewer_DtWk · 2025-10-28

**Soundness:** 3
**Presentation:** 3
**Contribution:** 3
**Rating:** 4
**Confidence:** 5

**Summary:**

This paper introduces CoT-PL, a novel framework that integrates visual chain-of-thought (CoT) reasoning into the pseudo-labeling process for open-vocabulary object detection (OVD). The method decomposes object understanding into three interpretable steps: region perception using SAM, zero-shot category recognition via MLLMs, and background grounding. It further proposes contrastive background learning (CBL) to disentangle object and background features. The authors demonstrate state-of-the-art performance on OV-COCO and OV-LVIS, with significant gains in challenging crowded and occluded scenes.

**Strengths:**

The idea of reformulating pseudo-labeling as a multi-step visual reasoning process is innovative and well-motivated. It effectively addresses key limitations of existing VLM-based methods (noisy boxes, caption dependency, background collapse).

The method achieves impressive gains on standard OVD benchmarks (+7.7 AP₅₀ on OV-COCO, +2.9 mask AP on LVIS for novel classes), with particularly large improvements in crowded and occluded settings.

The paper includes thorough ablation studies (e.g., CoT steps, MLLM variants, preprocessing strategies) that clearly demonstrate the contribution of each component.

**Weaknesses:**

Dependence on MLLM Quality: The method’s performance is tied to the zero-shot reasoning ability of the MLLM. Weaker models lead to lower-quality pseudo-labels, which may limit generalizability.

Computational Cost: The reliance on SAM and large MLLMs (e.g., Qwen2-7B) for pseudo-labeling makes the training pipeline computationally expensive and inference time relatively longer.

**Questions:**

CoT-PL relies on MLLM for multimodal reasoning, which could significantly increases its runtime compared to the baseline methods. To clarify the practical cost of CoT-PL, could you provide metrics on the pseudo-labeling speed (e.g., images/second) and the total GPU hours required to generate labels for a dataset like COCO?

Could the method be adapted to use smaller or more efficient MLLMs without significant performance drop? Have you explored model distillation or lightweight reasoning modules?

---

> ### Author Response · Authors · 2025-11-21
> **Response to Reviewer DtWk (Part 1/2)**
>
> We sincerely appreciate the reviewer’s careful feedback. We respond to your feedback below.
>
> > **[Q1] Dependence on MLLM Quality.**
>
> We conduct experiments with three open-source MLLMs of varying sizes (Tables 7–9) that are practical for real-world use. **The results show that performance does not degrade; instead, it even improves with a lighter model** such as BLIP2-2.7B, which belongs to the “small” category. Moreover, pseudo-label quality consistently improves with stronger MLLMs (Table 8), so our algorithm will naturally benefit as MLLMs advance (see Line 484). **Importantly, our method does not depend on high-end models (e.g., GPT-4o), although such models could provide further gains.**
>
> This robustness to lightweight MLLMs is largely explained by our image preprocessing and post-processing design within the CoT pipeline (Table 10 and Section H.1). Our image preprocessing has an even stronger effect on pseudo-label quality than simply changing the backbone MLLM. In practice, this means that **even with lightweight MLLMs, our image pre- and post-processing can still substantially improve the quality of pseudo-labels**. These observations are consistent with findings on related techniques (e.g., prompt engineering and in-context learning) in the broader LLM literature [1,2], where even high-end models (e.g., GPT-4) obtain additional, model-agnostic performance gains through test-time pre- and post-processing.
>
> `[Summary]` Even with lightweight MLLMs, combining our pre- and post-processing strategies with a CoT-inspired multi-step visual reasoning process can effectively produce high-quality pseudo-labels in practice.
>
> > **[Q2] Computational Cost.**
> >
> > **[Q3] Practical cost of CoT-PL pseudo-labeling**
>
> In response to reviewer s4bU’s Q3 and Q6, we conducted additional experiments comparing hardware requirements and pseudo-labeling time between our CoT pipeline and existing self-training-based pseudo-labeling methods. For COCO (~120k training images), our method uses 12 A6000 GPUs, each processing 10k images in parallel. The average pseudo-labeling time per image is approximately 15 seconds (±5s), and each GPU requires roughly 48 hours to process its 10k images. **Despite our superior performance, (1) the total cost remains comparable to the minimum time** (about 72 hours) required by conventional OVD pseudo-labeling pipelines that perform iterative self-training with a heavy student detector (NOTE THAT transformer-based student models require a huge amount of pseudo-labeling time).
>
> In terms of cost-effectiveness, a critical drawback of self-training-based OVD methods is that they rely solely on a student model, which **fundamentally lacks the capability to predict reliable pseudo-labels for small or occluded objects**. Moreover, additional self-training rounds can further degrade pseudo-label quality due to model overfitting [3]. In contrast, our method combines SAM and an MLLM in a CoT-based pipeline, generating **(2) detector-independent pseudo-labels that are not limited by the student model’s recall**. As validated in Table 4, this yields more effective pseudo-labeling in challenging scenes, offering clear advantages in both label quality and cost efficiency.
>
> We also emphasize that **(3) our pseudo-labeling cost is incurred only once in an offline stage before detector training**; at inference time, CoT-PL uses the same detector architecture as standard OVD models, so test-time cost remains unchanged.
>
> Moreover, Supplementary Section G and Figure 5 show that, under the same detector architecture, **(4) CoT-PL reduces overall training time by up to 4× compared to the baseline**.
>
> These **(1–4) results** further strengthen the efficiency advantage of our method over existing OVD pseudo-labeling approaches.
>
> ----
>
> **[Reference]**
>
> [1] Wang, Xuezhi, et al. "Self-Consistency Improves Chain of Thought Reasoning in Language Models." arXiv preprint arXiv:2203.11171 (2022).
>
> [2] Madaan, Aman, et al. "Self-Refine: Iterative Refinement with Self-Feedback." arXiv preprint arXiv:2303.17651 (2023).
>
> [3] Zhao, Shiyu, et al. "Taming Self-Training for Open-Vocabulary Object Detection." Proceedings of the IEEE/CVF Conference on Computer Vision and Pattern Recognition (CVPR) (2024).

---

> ### Author Response · Authors · 2025-11-28
> **Response to Reviewer DtWk (Part 2/2)**
>
> > **[Q4] Extension to smaller or more efficient MLLMs.**
>
> We observe a +3.7 AP50 improvement on novel classes even when pseudo-labels are generated using the lightweight BLIP2-2.7B model (Table 8), which belongs to the "small" category in the MLLM family. This is enabled by our semantic-anchor post-processing, which removes diffused class names from high-uncertainty predictions and keeps only reliable labels, regardless of the underlying MLLM’s capacity. As a result, **our CoT-based design can be safely applied even with lightweight MLLMs.**
>
> Since this question closely overlaps with Q1, we encourage the reviewers to also refer to our response to Q1.
>
> > **[Q5] Have you explored model distillation or lightweight reasoning modules?**
>
> Regarding distillation, our method basically distills CLIP knowledge into a student model during training, as shown in Figure 4. Advancing further, we also provided additional experimental results for reviewer hJNK’s Q1, showing that our model achieves even better SOTA performance by distilling contextual knowledge from the pseudo-captions.
>
> Most importantly, **even in its current form, our method is already highly cost-effective** (see our response to Q3 and Q4). Despite requiring almost the same pseudo-label generation time as prior OVD pseudo-labeling methods, it produces significantly more robust and higher-quality pseudo-labels. This demonstrates that our approach does not incur prohibitive computational cost, which we hope alleviates your concern about efficiency.
>
> Even so, we expect that exploring more efficient reasoning designs would be a promising direction for future work. One suggestion is to **replace the heavy segmentation foundation model SAM, which dominates the pseudo-label inference time per image, with a lightweight class-agnostic proposal generator**. We observe that such a lightweight generator is generally approximately 7× faster during inference than the heavy SAM, while causing only a slight performance drop, as validated in Table 7. Therefore, if efficiency is a primary concern, this replacement is a reasonable choice.
>
> `[Revision]` We will revise the manuscript to compare the inference speed of different proposal generators in Table 7, thereby offering a broader range of efficient detector choices.
>
> ----
>
> Thanks to the diverse feedback from all reviewers, we were able to prepare a more comprehensive response and additional experimental results, which you may also find informative. We especially appreciate the comments highlighting important points to clarify or revise. We hope these answers satisfactorily address your questions and concerns; we would be happy to further clarify.

---

### Official Review · Reviewer_hJNK · 2025-10-28

**Soundness:** 2
**Presentation:** 3
**Contribution:** 2
**Rating:** 6
**Confidence:** 4

**Summary:**

This paper proposes a pipeline (which is called a chain of thought in this paper) to utilize large vision-language models to produce pseudo labels for training open-vocabulary object detectors. The pipeline consists of three steps: first, generate object proposals by SAM and validate by MLLMs; second, generate free-form classnames for proposals containing an object; third, separate objects from background to encourage object–background disentanglement in training. The proposed pseudo-labeling method mitigates the limitations of region matching from CLIP and some missing objects in coarse captions. Training with improved pseudo labels, the model outperforms the baseline.

**Strengths:**

1. The proposed pseudo-labeling method can generate pseudo labels for all objects in the image, greatly eliminating the missing annotations.
2. The proposed pseudo-labeling method undergoes a rigorous pipeline to validate the quality of proposals, ensuring proposals contain the whole objects.
3. The proposed method takes background collapse into account, encouraging better object–background disentanglement.

**Weaknesses:**

1. There are also many improved pseudo-labeling methods. Comparing with them is necessary. For example:
- To tackle the **noisy pseudo boxes** problem, we can use a well-trained / self-trained open-vocabulary object detector as the labeler instead of the CLIP.
- To tackle the **caption dependency** problem, we can use an MLLM to rewrite the caption to get a more detailed one.
- Moreover, [1] also proposes a pseudo-labeling method using captions and achieves superior performance.

2. Comparing with generating pseudo labels from captions, the proposed method, which generates free-from classnames, has some problems that should be taken into account:
- Captions are annotated by humans, and the nouns in captions truly exist in the image. While MLLMs may have hallucinations. How to mitigate these noises.
- The nouns in the caption are a closed set. For example, all the people in the single image will be annotated as the same label 'person'. But the proposed method generates free-form labels and may generate 'person' and 'man' for different instances of the same class. How to handle the problem caused by synonyms and superclasses?

3. There are also many MLLMs with great region perception ability [2, 3]. Generating pseudo labels with them may get better performance and may eliminate preprocessing steps.
4. As shown in Table 6, filtering noisy and uncertain predictions will get higher performance. But why is the filtering not applied to OV-LVIS (threshold is set to 1 as shown in Line 924) ?



[1] A Hierarchical Semantic Distillation Framework for Open-Vocabulary Object Detection. In TMM 2025.

[2] Describe Anything: Detailed Localized Image and Video Captioning. In ICCV 2025.

[3] The All-Seeing Project: Towards Panoptic Visual Recognition and Understanding of the Open World. In ICLR 2024.

**Questions:**

Please see the weaknesses above.

---

> ### Author Response · Authors · 2025-11-21
>
> We sincerely appreciate the reviewer’s careful feedback. We respond to your feedback below.
>
> > **[Q1] Replacing CLIP with a well-trained OVD as the labeler.**
>
> As noted around Line 78, most OVD models are **themselves CLIP-based**, so using a “well-trained OVD labeler” still suffers from the same **direct CLIP image–text similarity limitations** (Figure 2). Moreover, such models struggle with long-tailed classes, including small or partially occluded objects (Table 4). This suggests that simply relying on well-trained OVD models as labelers cannot fundamentally resolve the issues discussed in the paper. In this context, our method restructures OVD into a three-stage CoT-based reasoning process and incorporates CBL to handle long-tailed objects better, a more reliable OVD framework.
>
> > **[Q2] MLLM-based caption rewriting.**
>
> This paper does **not assume the availability or reliability of captions**; instead, it focuses on developing a technique that performs effective pseudo-labeling purely from the input image itself. Our view is that if one can generate sufficiently high-quality pseudo-labels without relying on image captions, there is no strong justification for introducing the additional complexity of regenerating or refining captions via MLLMs. Such caption-based pipelines inevitably incur extra preprocessing cost and storage overhead. Therefore, our method is intentionally oriented toward “image-only OVD pseudo-labeling” in the long term, a more general OVD training framework.
>
> > **[Q3] Caption-based pseudo-labeling in [1].**
>
> Existing OVD methods can be roughly divided into two groups: (1) pseudo-labeling and (2) knowledge distillation. Our method belongs to (1), while [1] and BARON [2] belong to (2): they encode contextual information from captions into CLIP text embeddings and distill it into a student via a distillation loss. Our baseline, BARON [2], improves AP50 from 34.0 to 42.7 under this protocol, and [1] follows the same setup. For a fair comparison, following BARON [2], we apply the same caption-based distillation scheme to our method, which further improves performance from 41.7 AP50 to 50.4 AP50, as shown in the table below.
>
> **[Table 1]** OVD performance of our method with caption-based distillation.
>
> | Method              | Backbone | Detector    | $AP^N_{50}$ |
> |---------------------|----------|-------------|------------:|
> | BARON w/ KD [2]      | RN50 | Faster-RCNN | 42.7        |
> | HD-OVD [1] w/ KD    | RN50 | Faster-RCNN | 46.3        |
> | CoT-PL w/ KD (Ours) | RN50 | Faster-RCNN | **50.4**    |
>
> > **[Q4] Hallucination noise in MLLMs.**
> >
> > **[Q5] Synonyms and superclass ambiguity.**
>
> We already recognize that, like VLMs, MLLMs remain prone to hallucination. To mitigate this, we presented **several concrete remedies**:
> - **First**, for highly uncertain predictions, we allow the MLLM to answer “Unsure” safely instead of forcing a potentially wrong label.
> - **Second**, we apply a semantic-anchor post-processing step to the final pseudo-labels. Specifically, Table 6 shows that discarding classes whose pseudo-label counts fall below a certain threshold—thus treating them as unreliable—leads to improved label quality. This is motivated by the observation that, under high uncertainty, hallucinated free-form expressions tend to appear sparsely and in a highly diffused manner.
> - **Third**, we restrict the output space by limiting class names to 1–2 words and adding a few in-context examples in the prompt, as detailed in Supplementary Section J, preventing the model from drifting into uncontrolled free-form phrases.
>
> > **[Q6] Necessity of preprocessing steps.**
>
> As detailed in Supplementary Section H.1, our pre- and post-processing modules are standalone components that enhance MLLM predictions regardless of MLLM's strength. We use image preprocessing to help the MLLM see regions more clearly and semantic-anchor post-processing to keep only reliable pseudo-labels. As shown in Tables 6 and 10, removing these modules consistently harms performance, indicating that they are crucial for strong region-level perception.
>
> > **[Q7] Why is the threshold set to 1 for OV-LVIS?**
>
> We observe that annotation distributions differ greatly across datasets. In OV-LVIS, the number of classes is much larger than in OV-COCO (80 → 1.2K), and per-class annotations are much sparser, so the minimum base-class count is 1. As shown in Figure 3, our semantic-anchor filtering removes 1,321 unreliable classes with ≤1 annotation in OV-LVIS, leaving only 115 reliable classes (Table 2). Without this filtering, the model would be trained on numerous noisy labels, leading to suboptimal performance (Table 6).
>
> **[Reference]**
>
> [1] A Hierarchical Semantic Distillation Framework for Open-Vocabulary Object Detection. In TMM 2025.
>
> [2] Wu, Size, et al. "Aligning Bag of Regions for Open-Vocabulary Object Detection." Proceedings of the IEEE/CVF Conference on Computer Vision and Pattern Recognition (CVPR) (2023).

---

> ### Comment · Reviewer_hJNK · 2025-11-26
> **Reply to the comment**
>
> Thanks authors for providing the rebuttal. The results show that the proposed pseudo-labeling method is complementary to knowledge distillation and gets a higher performance. But (1) the proposed method is a labeling pipeline with some verification not a CoT-based reasoning process, which seems to be an overclaim. (Reviewer eugA shares the same comment.) (2) The assumption 'not assume the availability or reliability of captions' is not convincing as the pipeline uses MLLMs to generate object labels, which can also be used for generating captions.

---

> ### Author Response · Authors · 2025-11-26
>
> We sincerely thank the reviewer for taking the time to provide this additional comment and for the thoughtful clarification.
>
> > **[Q1] Regarding the CoT claim.**
>
> We appreciate the concern that the phrase “CoT-based reasoning” can be vague and may sound overclaimed. CoT-PL consists of (1) a pseudo-labeling pipeline and (2) a training scheme (CBL). Independent of (2), our intention in emphasizing “CoT” is to highlight that pseudo-label generation itself is reformulated as a CoT-inspired multi-step visual reasoning process, rather than as the single-shot VLM-based image–text matching used in existing pseudo-labeling methods. Specifically, CoT-PL decomposes pseudo-labeling into a three-stage, CoT-inspired pipeline, where each stage targets one of the three aforementioned OVD challenges described in Line 80: region selection (noisy boxes), category verification (caption dependency), and background grounding (background collapse). Each stage takes as input not only the image but also the intermediate decisions from previous stages (which proposals are retained, which candidate classes are filtered, how context is recognized). **This is what we mean by a “chain-of-thought” formulation in our setting: pseudo-labels are obtained by decomposing the complex scene-understanding problem into a sequence of intermediate visual decisions**. As shown in Table 5, both replacing our pipeline with such direct matching baselines and collapsing the three stages into a one-step pseudo-labeler lead to a clear performance drop, especially on crowded and occluded subsets, which indicates that the multi-step CoT-inspired pseudo-labeling structure itself is important for OVD.
>
> `[Revision]` To address the reviewer’s concern and avoid over-claiming, we will carefully revise the wording in the manuscript to consistently describe our pseudo-labeling process as a “CoT-inspired multi-step visual reasoning process”, rather than using the more generic phrase “CoT-based reasoning process.”
>
> > **[Q2] Regarding caption assumptions.**
>
> We appreciate the reviewer’s point and acknowledge that our original statement (“does not assume the availability or reliability of captions”) was phrased too strongly. We agree that MLLMs could, in principle, also be used to generate captions, which can then be used to reason about potential novel-class labels, as in existing pseudo-labeling methods (Lines 72–76).
>
> Our intention, however, was to question whether one must go through the **cumbersome caption-generation process** in the first place. Existing caption-based OVD pseudo-labeling methods must first obtain, process, or regenerate captions, which inevitably introduces **extra sophisticated preprocessing steps to derive pseudo-labels from captions** and **additional storage overhead for caption supervision**. In contrast, our design **does not inherently depend on captions for efficient pseudo-label generation**: all pseudo-labels are automatically generated directly from images and region proposals through a CoT-inspired multi-step pseudo-labeling pipeline, leveraging the zero-shot capability of MLLMs (Lines 97–98); we never read, refine, or store either expensive human-annotated captions or external caption corpora. Despite this increased efficiency, CoT-PL not only produces higher-quality pseudo-labels than existing methods in complex scenes (Table 4), but also achieves state-of-the-art OVD performance when trained with these labels (Tables 1 and 3).
>
> `[Revision]` We are grateful for this opportunity to clarify our intention and will revise the manuscript accordingly by (1) explicitly contrasting caption-dependent pipelines with our caption-free, CoT-inspired pseudo-labeling formulation, and (2) rephrasing our assumption more precisely as “does not require human-annotated captions or external caption corpora,” which we believe better reflects our goal of providing a more efficient, real-world OVD pseudo-labeling paradigm where only input images are available.
>
> ----
>
> We sincerely appreciate your thoughtful feedback and hope these answers satisfactorily address your questions and concerns. We would always be happy to provide any further clarification you may need!

---

> ### Comment · Reviewer_hJNK · 2025-11-28
> **Reply to the comment**
>
> Thanks authors for providing further clarifications.

---

> ### Author Response · Authors · 2025-11-28
>
> Dear Reviewer hJNK,
>
> We sincerely appreciate the time and effort you have dedicated to reviewing our work and providing such valuable feedback. We are very pleased that our response has fully resolved all of the questions and concerns you raised. The clarifications that we have developed based on our discussion also help to address similar questions raised by the other reviewers. In this regard, we are deeply grateful that your comments have contributed to a more constructive and engaging discussion within the community.
>
> Please do not hesitate to let us know if any further clarification or additional information would be helpful.
>
> Best regards,
>
> Authors

---

### Official Review · Reviewer_s4bU · 2025-10-28

**Soundness:** 3
**Presentation:** 4
**Contribution:** 3
**Rating:** 6
**Confidence:** 4

**Summary:**

This paper proposes CoT-PL, a new framework for pseudo-labeling in open-vocabulary object detection (OVD).
Unlike prior methods that rely on direct image–text alignment using vision-language models (VLMs), CoT-PL formulates pseudo-label generation as a multi-step visual reasoning process—a “chain-of-thought” approach for visual understanding.

The framework explicitly tackles three key weaknesses of current OVD pseudo-labeling methods:

1. Noisy pseudo boxes, caused by co-occurrence bias in image-level VLM supervision.

2. Caption dependency, where objects missing from captions remain unlabeled.

3. Background collapse, where occluded or unlabeled instances are incorrectly learned as background.

To address these, CoT-PL integrates three reasoning stages:

1. Region Perception: SAM generates candidate masks, and an MLLM verifies object existence to remove spurious boxes.

2. Category Recognition: A zero-shot reasoning module assigns labels to each region without relying on captions.

3. Background Grounding: Contrastive Background Learning (CBL) separates unlabeled background from true objects by using grounded background cues as negative training signals.

The method achieves consistent improvements on OV-COCO and OV-LVIS benchmarks, producing higher-quality pseudo labels and more robust detectors.

**Strengths:**

- Clear problem identification: The paper clearly articulates three major weaknesses of existing OVD pseudo-labeling pipelines (noisy boxes, caption dependency, background collapse) and provides a coherent reasoning-based solution.

- Conceptual originality: Reinterpreting pseudo-labeling as a structured reasoning process is both novel and well motivated.

- Unified and interpretable design: The integration of SAM, MLLM reasoning, and contrastive background learning forms a consistent, interpretable pipeline that can be easily understood and reproduced.

- Strong empirical results: The proposed approach improves pseudo-label quality and achieves competitive or superior performance under complex, occluded conditions.

**Weaknesses:**

- Limited comparison with latest baselines: The paper does not include direct comparisons with very recent (2024–2025) state-of-the-art OVD models, which makes it difficult to fully gauge its competitiveness.

- Backbone limitation: Most experiments rely on ResNet-50, which may not capture the performance trends of newer backbones (e.g., ViT, Swin).

- Efficiency and scalability: The computational cost of the multi-step reasoning pipeline (especially SAM and MLLM inference) is not analyzed, raising concerns about scalability to large-scale or real-time settings.

- Dataset scope: Evaluation is limited to COCO and LVIS. Broader tests on real-world or open-set data would strengthen claims about generalization.

**Questions:**

- How does CoT-PL perform when compared directly with the most recent SOTA OVD models?

- What is the runtime overhead introduced by the SAM + MLLM reasoning process? Could the authors discuss the computational trade-offs?

- How sensitive is CoT-PL to the choice and scale of the multimodal language model used?

- Could the chain-of-thought reasoning be extended to temporal or video-based object detection tasks?

---

> ### Author Response · Authors · 2025-11-21
> **Response to Reviewer s4bU (Part 1/2)**
>
> We sincerely appreciate the reviewer’s careful feedback. We respond to your feedback below.
>
> > **[Q1, Q5] Comparison with recent SOTA OVD models.**
>
> We provide updated performance comparisons against the latest OVOD models proposed after 2024, including those introduced in this paper. The table shows that **our model continues to achieve state-of-the-art performance on novel classes in the OVD setting.**
>
> **[Table 1]** Comparison of the latest baselines on the OV-COCO validation set.
> | Method         | Backbone | $AP^{N}_{50}$ |
> |----------------|----------|------------:|
> | OV-DQUO     | RN50     | 39.2        |
> | CCKT-Det    | RN50     | 38.0        |
> | OC-OVD      | RN50     | 36.6        |
> | LBP         | RN50     | 37.8        |
> | CAKE [1]        | RN50     | 39.1        |
> | ATAS [2]          | ViT-B    | 37.2        |
> | CoT-PL (Ours) | RN50     | **41.7**    |
>
> > **[Q2] Backbone limitation.**
>
> We report additional results in which our model is trained with a heavier and more powerful RN50x4 backbone, which is comparable in capacity to Swin and ViT. Across these configurations, **our method consistently maintains state-of-the-art performance, demonstrating that it scales well with larger backbones and exhibits strong scalability in OVD scenarios.**
>
> **[Table 2]** Comparison with stronger backbones on the OV-COCO validation set.
>
> | Method         | Backbone | $AP^N_{50}$ |
> |----------------|----------|------------:|
> | OV-DQUO     | RN50x4   | 45.6        |
> | CLIPSelf    | ViT-L    | 44.3        |
> | CORA        | RN50x4   | 41.7        |
> | BIND        | ViT-L    | 41.5        |
> | CCKT-Det    | Swin-B   | 41.9        |
> | CoT-PL (Ours)  | RN50x4   | **47.1**    |
>
> > **[Q3] Efficiency and scalability.**
> >
> > **[Q6] Runtime overhead of the SAM + MLLM reasoning.**
>
> On the COCO training set (~120k images), our pseudo-labeling takes about 48 hours on 12×A6000 GPUs (≈ 576 GPU-hours). In comparison, typical OVD pseudo-labeling pipelines require at least three rounds of self-training [3], each taking approximately one day on 8×A6000 GPUs (≈ 576 GPU-hours), and transformer-based variants [4] are likely even more expensive. Thus, our approach **has a comparable overall computational cost, but produces more robust and higher-quality pseudo-labels**, as shown in Table 4.
>
> We provide more detailed clarifications on efficiency in our responses to Q2–Q3 of Reviewer DtWk, which we believe are helpful for better understanding.
>
> > **[Q4] Dataset scope.**
>
> Supplementary Table 12 reports transfer learning results not only on the COCO / LVIS datasets but also on MS-COCO / Objects365. Such transfer settings are widely adopted in the OVD literature as a way to validate models under more open-set conditions. Under the same setup, **our method consistently achieves state-of-the-art performance, demonstrating robust OVD inference even when applied to heterogeneous datasets.**
>
> ----
>
> `[Revision]` These additional results further highlight the strength and competitiveness of CoT-PL, and we will incorporate the extended comparisons and analyses into the revised manuscript to better showcase these strengths.
>
> **[Reference]**
>
> [1] Ma, Qian, et al. "CAKE: Category Aware Knowledge Extraction for Open-Vocabulary Object Detection." Proceedings of the AAAI Conference on Artificial Intelligence 39.6 (2025): 5982–5990.
>
> [2] Yeo, Juan, et al. "ATAS: Any-to-Any Self-Distillation for Enhanced Open-Vocabulary Dense Prediction." Proceedings of the IEEE/CVF International Conference on Computer Vision (ICCV) (2025).
>
> [3] Zhao, Shiyu, et al. "Taming Self-Training for Open-Vocabulary Object Detection." Proceedings of the IEEE/CVF Conference on Computer Vision and Pattern Recognition (CVPR) (2024).
>
> [4] Wu, Xiaoshi, et al. "CORA: Adapting CLIP for Open-Vocabulary Detection with Region Prompting and Anchor Pre-Matching." arXiv preprint arXiv:2303.13076 (2023).

---

> ### Author Response · Authors · 2025-11-28
> **Response to Reviewer s4bU (Part 2/2)**
>
> > **[Q7] Sensitivity of CoT-PL to the multimodal language model.**
>
> Table 8 shows that CoT-PL benefits from scaling up the open-source MLLM: the improvement over the baseline increases from +3.6 AP50 with BLIP-2-2.7B (*small-sized MLLM*) to +7.7 AP50 with larger models such as Qwen2-7B (*medium-sized MLLM*). In particular, once we use a sufficiently strong model such as InstructBLIP-7B, replacing it with other 7B-scale MLLMs results in only small performance differences of less than 2 AP50, indicating that **CoT-PL is not highly sensitive to the exact choice of MLLM at that scale**. As even stronger MLLMs (e.g., GPT-4o [1]) continue to appear, this observation suggests that both pseudo-label quality and OVD performance consistently improve and stabilize at higher levels as the underlying MLLM becomes more capable.
>
> > **[Q8] Extension to temporal or video tasks**
>
> The proposed CoT-based module is **self-contained, enabling it to be extended from images to videos**. One possible direction is to reuse the original three-stage CoT process and **add a fourth stage that explicitly reasons over time**. At this stage, the model uses the intermediate evidence accumulated in the earlier CoT stages across frames to check whether boxes with slightly changing appearances correspond to the same object, and then aggregates this evidence over multiple frames to produce temporally consistent OV pseudo-labels with refined boxes and stable IDs. These labels can then be used to train video detectors with temporal consistency losses or ID stability constraints.
>
> Consequently, we extend the CoT process from a single box to a time-aligned sequence of boxes, which makes the video extension straightforward and practical.
>
> ----
>
> We hope these responses address your questions and concerns thoroughly; we are happy to provide further clarification if needed.
>
> **[Reference]**
>
> [1] OpenAI. "GPT-4o System Card." arXiv preprint arXiv:2410.21276 (2024).

---

### Official Review · Reviewer_eugA · 2025-10-30

**Soundness:** 3
**Presentation:** 3
**Contribution:** 2
**Rating:** 4
**Confidence:** 3

**Summary:**

This paper introduces a three-step pipeline—comprising pseudo-box generation, pseudo-label assignment, and background extraction—to improve pseudo-label quality for open-vocabulary object detection. The method demonstrates strong results on the OV-COCO and OV-LVIS benchmarks.

**Strengths:**

The paper is well-organized and easy to follow.

The proposed three-step pipeline is reasonable and shows potential for mitigating the long-tail problem in object detection.

The proposed method demonstrates strong performance on the OV-COCO and OV-LVIS benchmarks.

**Weaknesses:**

A fundamental question is: what is the difference between CoT based pseudo-boundary box generation and CoT-based bounding box prediction? If the goal is to use pseudo-annotations to train a more efficient model, then the long-tail issue of partial and small objects still exists.

Although the authors emphasize the use of CoT, the method functions more like a fixed, three-step pipeline than a dynamic reasoning process. A key limitation is its failure to reason about the contextual relationships between objects in a scene. For instance, instead of inferring that the partial object is a "person" because the object is on a "skateboard" (as seen in Figure 2a), the pseudo-label assignment module appears to process each object independently.

Consequently, the work is perceived more as a clever system optimization built upon existing methods, rather than a deep investigation into how fine-grained visual reasoning, such as Chain-of-Thought, can advance the field of open-vocabulary object detection.

**Questions:**

Please see the weakness.

---

> ### Author Response · Authors · 2025-11-21
> **Response to Reviewer eugA (Part 1/2)**
>
> We appreciate your response and the time you took to share your perspective. We respond to your feedback below.
>
> > **[Q1] CoT-based box prediction vs. CoT-based pseudo box generation.**
>
> (1) “CoT-based box prediction“ and (2) “CoT-based pseudo box generation“ are **fundamentally aimed at different goals**. In (1), boxes must be newly generated in a CoT-based manner for every new image, independently of model training and inference. This makes it challenging for the method to serve as a core detection approach in real-world environments, where vast numbers of images are present. In contrast, (2) is a training-only pseudo-annotation scheme that improves a model’s generalization for OV recognition. An OV detector trained with such pseudo-labels can effectively predict the bounding boxes of both base and novel objects on unseen images, making it suitable for deployment in real-world settings.
>
> > **[Q2] Long-tailed object recognition.**
>
> **(1) The proposed CoT-based method can produce pseudo-labels even for small or partially occluded objects**. However, although this provides some supervision for such cases, we agree that merely “using these pseudo-labels” does not, by itself, justify the claim that the model will implicitly learn to infer small or occluded objects robustly. We therefore propose **(2) CBL to ensure that the model indeed learns small and occluded objects.** CBL is motivated by the observation that such objects are often misclassified as part of the background class during training (Lines 88–89). It provides a contrastive learning signal, ensuring that long-tailed object instances are not misclassified as background but rather assigned to the nearest object class in the embedding space (Line 296).
>
> As a result, applying CBL yields a decent improvement of about +2.1 AP50 (Table 5). Through this **integrated (1)–(2) system**, we substantially improve detection performance for long-tailed object classes in both *explicit* (pseudo-label) and *implicit* (CBL) manners.
>
> > **[Q3] Contextual relationships among objects.**
>
> Our method **indeed incorporates contextual reasoning among objects through an image preprocessing pipeline for MLLM inference**, as detailed in Supplementary Section H.1. As shown in Figure 6, we construct image inputs by centering the target object while applying blur and grayscale transformations to background regions, **prompting the MLLM to reason about the target jointly with its surrounding context**. Line 246 provides evidence that MLLMs already exhibit strong multi-class recognition in multi-object scenes and can capture such contextual cues. Leveraging this contextual understanding, our design stabilizes inference of the target object even when it is partially visible (e.g., the “person” and “skateboard” in Figure 2a).
>
> Consistently, Table 10 in the supplementary material shows that including background context in the query yields higher-quality pseudo-labels than querying only the central individual object.
>
> `[Summary]` Our CoT-based chained reasoning complements prior pseudo-labeling methods that largely ignore context **by combining the MLLM’s contextual understanding with foundation-segmentation-based preprocessing.**

---

> ### Author Response · Authors · 2025-11-28
> **Response to Reviewer eugA (Part 2/2)**
>
> > **[Q4] Fixed vs. dynamic reasoning.**
>
> We understand and appreciate the reviewer’s concern that CoT-PL might seem like a fixed three-step pipeline. This impression probably comes from our manuscript focusing on explaining the flow of the CoT structure, without enough emphasis on the fact that these stages interact dynamically through their intermediate reasoning outputs.
>
> To clarify, our goal in emphasizing CoT for complex scene understanding is to show that, within this structure, pseudo-labeling is performed **through decisions that are dynamically chained across stages, with each stage explicitly conditioned on the previous stage’s intermediate outputs**, rather than through a direct VLM-based image–text matching as in earlier pseudo-labeling approaches. This process is briefly summarized in Lines 78–90:
> - **Stage 1 (region perception):** SAM proposes multiple regions, and the MLLM selects or rejects them, so the proposals that are kept adapt to each image.
> - **Stage 2 (category recognition):** The MLLM may generate multiple candidate class names with varying confidence; uncertain or rare labels are filtered using our semantic-anchor mechanism, so the final label set depends on the specific image and region.
> - **Stage 3 (background grounding):** which regions are deemed background anchors versus true objects is based on the distribution of pseudo-labels from earlier stages. Importantly, these background cues naturally support online CBL training, improving performance on long-tailed objects as discussed in Sections 3.3–3.4.
>
> Table 5 supports this view: collapsing the three stages into a single-step pseudo-labeler, or removing stages, causes noticeable performance drops, especially on crowded and occluded subsets. This demonstrates that dynamic interactions between intermediate outputs are crucial for robust pseudo-labeling.
>
> `[Summary]` The chained three stages **interact dynamically**: the proposals and labels decided in Stages 1 and 2 directly influence Stage 3, and subsequent decisions are explicitly conditioned on earlier ones rather than made in isolation.
>
> `[Revision]` We will revise the manuscript to better highlight the dynamic interaction among the three stages through their intermediate outputs and to describe CoT-PL more precisely as a “CoT-inspired multi-step visual reasoning process,” instead of using the more generic phrase “CoT-based reasoning process.”
>
> > **[Q5] Deep investigation of CoT for OVD**
>
> At a high level, we are **the first to explicitly define pseudo-labeling for OVD as a complex scene-understanding problem** with three main challenges (VLM-induced noisy boxes, caption dependency, and background collapse). We **break down this complex issue into a CoT-inspired three-stage visual reasoning process for pseudo-label generation** that directly addresses each challenge. Our conceptual contribution therefore extends beyond proposing a single pipeline: the three-stage CoT-inspired decomposition **illuminates where and how multi-step visual reasoning should be applied in OVD**. Each stage processes both the image and the intermediate decisions from earlier stages (which proposals are kept, which candidate classes are filtered, and how context is perceived), and these decisions are reused by later stages.
>
> Table 5 provides a detailed stage-wise ablation, comparing variants with and without each stage and collapsing the three stages into a one-step pseudo-labeler. The performance drops observed, especially in crowded and occluded scenes, confirm that the CoT-inspired three-stage structure is vital for OVD.
>
> We believe this formulation offers a deeper insight into how CoT-inspired multi-step visual reasoning benefits robust OVD pseudo-labeling, rather than simply adding another system-level optimization on top of existing methods.
>
> ----
>
> We hope these responses address your questions and concerns thoroughly; we are happy to provide further clarification if needed.

---

### Author Response · Authors · 2025-12-01
**Summary of Rebuttal Progress Following the Information Leak Incident**

Dear AC and Reviewers,

We are sorry to hear about the recent information leak on OpenReview and its impact on the ICLR reviewing process. We recognize that this unforeseen incident has created additional workloads, and we would like to express our sincere gratitude to the AC for the extra time and effort devoted under these difficult circumstances.

In an effort to help ease the AC’s burden, we provide below a concise summary of the main points of progress that were explicitly recognized by the reviewers.

> Addressing Reviewer **eugA**'s concerns **(Rating: 4, Confidence: 3)**:

- `[Q1] (1) CoT-based box prediction vs. (2) CoT-based pseudo box generation.`
We clarified that (1) is a costly per-image CoT inference scheme, whereas **(2) is a training-only pseudo-annotation method** that yields an OV detector deployable at scale.

- `[Q2] Long-tailed object recognition.`
We clarified that **the combination of CoT-based pseudo-labels and CBL effectively mitigates long-tailed recognition** for small and occluded objects.

- `[Q3] Contextual relationships among objects.`
CoT-PL **explicitly feeds surrounding context to the MLLM**, yielding higher-quality pseudo-labels than isolating each object.

- `[Q4] Fixed vs. dynamic reasoning.`
CoT-PL **engages in dynamically chained reasoning**, and comprehensive ablations demonstrate that this dynamic interaction is essential for performance.

- `[Q5] Deep investigation of CoT for OVD.`
We showed how CoT advances OVD **by decomposing hard scene-level pseudo-labeling into three easier, chained reasoning steps** that directly address the core OVD failure modes.

⇒ **eugA** mainly focused on **validity**, and we clarified that **the proposed modules are individually necessary** and that their integration consequently achieves new SOTA performance in OVD.

> Addressing Reviewer **s4bU**’s concerns **(Rating: 6, Confidence: 4)**:

- `[Q1, Q5] Comparison with recent SOTA OVD models.`
We added comparisons to post-2024 SOTA methods and confirmed that **CoT-PL remains SOTA.**

- `[Q2] Backbone limitation.`
We added stronger backbones, showing that **CoT-PL scales well while preserving its SOTA performance in this setup.**

- `[Q3, Q6] Efficiency.`
We showed that our CoT pipeline has a labeling cost **comparable to existing methods while yielding better labels.**

- `[Q4] Dataset scope.`
We presented cross-dataset transfer results demonstrating that **CoT-PL generalizes well under open-set conditions.**

- `[Q7] Sensitivity to the MLLM.`
We showed that performance is **stable across models of the same scale** while consistently improving with stronger MLLMs.

- `[Q8] Extension to temporal/video tasks.`
We outlined a natural extension where our CoT pipeline is augmented with **a temporal reasoning stage for video OVD.**

⇒ **s4bU** mainly focused on **performance**, and we directly addressed this by showing that **CoT-PL consistently achieves SOTA performance across diverse settings.**

> Addressing Reviewer **hJNK**’s concerns **(Rating: 6, Confidence: 4)**:

- `[Q1] Replacing CLIP with a well-trained OVD labeler.`
Existing OVD labelers inherit the same limitations as direct CLIP matching (Line 78), whereas **CoT-PL directly addresses them and offers a more robust alternative.**

- `[Q2, Q3] Caption rewriting and caption-based methods.`
We positioned CoT-PL as **caption-free** pseudo-labeling that is complementary to caption-based distillation, and showed that **combining them outperforms previous SOTA caption-based OVD baselines.**

- `[Q4, Q5] MLLM hallucination.`
We introduced **several uncertainty handling strategies and semantic-anchor filtering** to suppress hallucinations and stabilize free-form labels.

- `[Q6] Necessity of preprocessing with strong MLLMs.`
We showed that our preprocessing **consistently improves region-level perception, even with strong MLLMs.**

- `[Q7] Threshold choice on OV-LVIS.`
We justified the LVIS threshold **by its sparse, long-tail label distribution** and showed that it **effectively filters unreliable classes.**

⇒ **hJNK** mainly focused on **robustness**, and our unified system remains **reliably robust across the challenging settings** presented by the reviewer.

> Addressing Reviewer **DtWk**’s concerns **(Rating: 4, Confidence: 5)**:

- `[Q1, Q4] Dependence on MLLM quality across model scales.`
We clarified that **CoT-PL remains effective even with small MLLMs and further improves as MLLMs get stronger.**

- `[Q2, Q3] Computational cost.`
We clarified that CoT-PL is **highly cost-effective**, yielding better labels at comparable labeling cost and unchanged inference cost.

- `[Q5] Distillation and efficiency.`
We highlighted that **our framework already leverages CLIP distillation and can be further sped up with lighter proposal generators.**

⇒ **DtWk** mainly focused on **efficiency**, and we convincingly addressed these concerns by showing that **CoT-PL is highly cost-effective** in OVD pseudo-labeling compared to existing methods.

---

### Meta-Review · Area_Chair_7kRi · 2025-12-26

**Summary:**

The paper proposes a three-step pipeline—comprising pseudo-box generation, pseudo-label assignment, and background extraction—to improve pseudo-label quality for open-vocabulary object detection. However, reviewers question whether the approach offers genuine reasoning advances beyond a fixed system pipeline and sufficiently differentiates from existing pseudo-labeling methods.
Considering the reviewers’ concerns, we regret that the paper cannot be recommended for acceptance at this time. The authors are encouraged to consider the reviewers’ comments when revising the paper for submission elsewhere.

**Reviewer Concerns:**

Key concerns include (1) unclear distinction between CoT based pseudo-boundary box generation and CoT-based bounding box prediction, (2) limited contextual object reasoning, (3) reliance on MLLM Quality, (4) computational cost.

**Reviewer Scores:**

Reviewers rate soundness and presentation mostly good, but contribution ranges from fair to good. Ratings span marginally below to marginally above acceptance, reflecting mixed confidence. While strengths and results are acknowledged, concerns about novelty depth, comparisons, and practicality motivate an overall reject recommendation.

---

### Decision · Program_Chairs · 2026-01-26

Reject